# Conformational maps of human 20S proteasomes reveal PA28- and immuno-dependent inter-ring crosstalks

Jean Lesne [1,3,5], Marie Locard-Paulet [1,2,5], Julien Parra [1], Dušan Zivković [1], Thomas Menneteau [1,4], Marie-Pierre Bousquet[1], Odile Burlet-Schiltz[1] & Julien Marcoux [1✉]

Hydrogen-Deuterium eXchange coupled to Mass Spectrometry (HDX-MS) is now common practice in structural biology. However, it is most of the time applied to rather small oligomeric complexes. Here, we report on the use of HDX-MS to investigate conformational differences between the human standard 20S (std20S) and immuno 20S (i20s) proteasomes alone or in complex with PA28αβ or PA28γ activators. Their solvent accessibility is analyzed through a dedicated bioinformatic pipeline including stringent statistical analysis and 3D visualization. These data confirm the existence of allosteric differences between the std20S and i20S at the surface of the α-ring triggered from inside the catalytic β-ring. Additionally, binding of the PA28 regulators to the 20S proteasomes modify solvent accessibility due to conformational changes of the β-rings. This work is not only a proof-of-concept that HDX-MS can be used to get structural insights on large multi-protein complexes in solution, it also demonstrates that the binding of the std20S or i20S subtype to any of its PA28 activator triggers allosteric changes that are specific to this 20S/PA28 pair.

[1] Institut de Pharmacologie et de Biologie Structurale (IPBS), Université de Toulouse, CNRS, UPS, Toulouse, France. [2] Novo Nordisk Foundation Center for Protein Research, University of Copenhagen, Copenhagen, Denmark. [3] Present address: Centre de Biologie Structurale, CNRS, Université de Montpellier, INSERM, 34090 Montpellier, France. [4] Present address: Institute of Structural and Molecular Biology, Division of Biosciences, University College London, London WC1E 6BT, UK. [5] These authors contributed equally: Jean Lesne, Marie Locard-Paulet. ✉email: julien.marcoux@ipbs.fr

The ubiquitin proteasome system (UPS) is central to proteostasis. Its most downstream element is the 26S proteasome that clears the cells of abnormal, denatured, or damaged proteins and regulates degradation of short-lived proteins, in most cases conjugated to ubiquitin. The 26S proteasome is a highly conserved compartmentalized multicatalytic protease composed of more than 30 subunits constituting the 20S catalytic core and the 19S regulatory complex[1]. Its activity is directly involved in many cytokines and hub proteins intracellular concentration[2], and regulates immunogenic peptide production[3]. It is the focus of intense regulation and dynamically localizes within cells in response to physiological and external perturbations[4,5] and its genetic polymorphism is directly responsible for numerous pathologies including cancer, heart disease, and type 2 diabetes to name a few[6,7].

The main mechanism regulating the proteasome activity is the substitution of the catalytic subunits β1, β2, and β5, respectively harboring caspase-like, trypsin-like, and chymotrypsin-like activities, and constituting the standard 20S (std20S) with other subunits. For example, the immunoproteasome (i20S) contains the immuno-subunits β1i, β2i, and β5i that have different cleavage specificities and generate immunogenic peptides[8]. It is strongly expressed in immune cells and can be induced in most tissues by interferon-γ.

The proteolytic activity of the 20S is also regulated by interacting proteins, the most common being the ATP-dependent 19S activator involved in the UPS degradation pathway. However, less abundant ubiquitin-independent regulators such as PA28αβ, PA28γ, or PA200 also activate the 20S and modify its substrate specificity. In addition, a vast collection of Proteasome Interacting Proteins (PIPs) regulate or assist proteasomal functions[9–16].

Besides its subunit composition and association with regulators and/or PIPs, the proteasome is subjected to a wide variety of post-translational modifications (PTMs) affecting its subunits activity[17–19]. The combination of these three levels of regulation (catalytic subunits, regulators, PTMs) implies a wide proteasome subtype heterogeneity, which probably results in specialized functions that can adapt protein degradation pathways to changing conditions in the cell. One of today's main challenges is to understand how different proteasome subtypes influence its proteolytic activity and the cell function.

The proteasome has been intensely studied from a structural and functional point of view since its discovery in 1988[20]. Electron microscopy (EM) allowed to observe the 20S in complex with different regulators[21–26] together with certain catalytic intermediate-states of the 26S[27]. Covalent cross-linking coupled to mass spectrometry (MS) helped refining 26S structures[28,29] and generating structural models involving different partners, including the 19S[23], Ecm29[15], Ubp6[30], and other PIPs[15]. X-ray crystallography also provided high-resolution structures of proteasomes alone or in the presence of covalent inhibitors[31], regulators[32,33], and complexes of proteasome with their associated regulators[15,31,32].

The complex size is obviously a limit for its analysis by Nuclear Magnetic Resonance (NMR). However, the structure of the eukaryotic proteasome from *Thermoplasma acidophilum*, containing only one type of subunit α and β, has been resolved. It showed evidence of a remodelling of the catalytic sites located in the center of the proteasome upon binding of a regulatory particle on its surface, which can be described as an outer-to-inner allosteric change[34]. A reverse inner-to-outer change was also observed at the binding interface when modifying the catalytic site, but this was not confirmed by the comparison of the std20S X-ray structure[35] with the recent i20S X-ray[36] and cryo-EM[37] structures (RMSD = 0.392 and 0.480 Å, respectively). However, the outer-to-inner mechanism was recently shown in the human

20S cryo-EM structure upon PA200 binding[25]: conformational changes were found not only at the binding interface (opening of the pore), but also down to the catalytic sites.

Despite these achievements, high-resolution methods are still limited by either the size, the heterogeneity, or the dynamics of the purified complexes. This is evidenced by the small number of structures of the 20S bound to regulators and their numerous unresolved regions.

Here, we optimized the emerging method Hydrogen-Deuterium eXchange coupled to MS (HDX-MS) to investigate conformational changes occurring upon binding of the std/i20S to PA28 regulators. This method, developed in the 90 s, enables the detection of conformational differences between two protein samples. Briefly, the proteins or protein complexes of interest are diluted in a deuterated buffer and hydrogen atoms from the peptide bonds are exchanged with the deuterium atoms of the solution. The rate of exchange of each amide hydrogen depends on its solvent accessibility and involvement in the stabilization of secondary structures: regions that are relatively more solvent-accessible or flexible present a higher deuteration rate than the ones that are hidden in the core of the protein or rigid. This expanding method can thus be used to study protein dynamics[38], compare protein conformations (in different buffer conditions, after mutation, ligand binding)[39], or to identify protein–protein or protein–ligand binding interfaces[40,41]. We present the HDX-MS analysis of the entire human std and i20S core particles as well as their PA28αβ and PA28γ regulators, mapping solvent accessibility/dynamics for each complex. The conformational maps of the std and i20S present significant differences, and their differential analysis with and without regulators provide a molecular rationale for their distinct functions. Our data provide evidence for the human 20S inner-to-outer allosteric change upon incorporation of the immuno-subunits and the reverse outer-to-inner transduction signal upon PA28 binding, illustrating the interplay between the different proteasome regulation pathways. Altogether, this work highlights the potential of HDX-MS to generate low resolution but informative structural information on large hetero-oligomeric complexes. It opens the door to many other applications, including identifying PIPs binding surfaces and the stabilizing/destabilizing effects of other regulators of the 20S activity, including small molecules.

## Results

**Dynamic interfaces uniting the four rings of the 20S proteasome.** HDX consists in the exchange of backbone amide hydrogens (H) with deuteriums (D) in proteins in solution. The exchange rate depends on the solvent accessibility and/or the flexibility of a given region, thereby providing low resolution but crucial information on protein conformation. We performed two distinct analyses (Fig. 1a): (i) identification of the most accessible and/or dynamic regions of the proteins in solution, and (ii) differential analysis of each protein alone or in complex. We monitored the incorporation of deuterium for each sample from 0 to 30 min in triplicate experiments (Fig. 1b). For differential analyses, we compared the peptide relative deuterium uptakes (RDU) of the proteins alone vs. in presence of their binding partner. The peptides were considered significantly different when they presented a $P$ value of the ANOVA $\leq 0.01$ after correction for multiple testing (Benjamini–Hochberg) and three successive time points with an absolute difference between the two conditions >4 times the experimental standard deviation observed in the dataset (Fig. 1c) (see Methods for more detailed information). The peptide-level RDUs or differences of RDU were mapped to the protein sequences as described in Fig. 1d: when several peptides covered the same region of a protein we

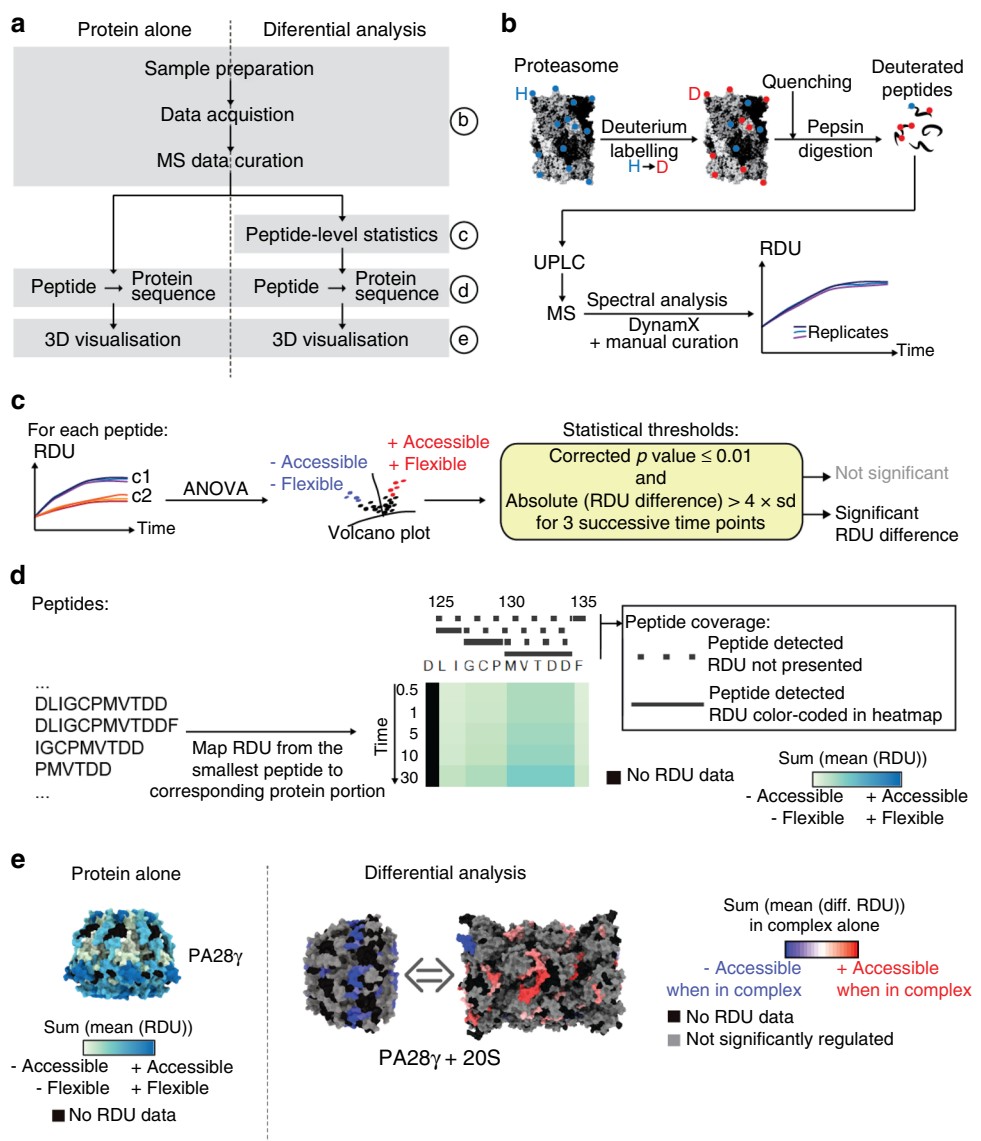

**Fig. 1 HDX-MS workflow applied to the analysis of the 20S proteasome and its regulators. a** Overview of the workflows applied to the analysis of proteins alone (left) or differential analysis of proteins alone vs. in complex (right). The different steps are described in more detail in **b–e** as referred to on the right side. **b** Sample preparation and MS data acquisition. Proteins (alone or in complex) were subjected to hydrogen (blue points) to deuterium (red points) exchange in triplicate experiments for 0, 0.5, 1, 5, 10, and 30 min before being subjected to pepsin digestion. Peptides were then separated by liquid chromatography (UPLC) and analysed by MS. Spectra were curated using DynamX before statistical analysis and/or mapping of their RDU to the protein sequence (consolidation). **c** Statistical comparison of proteins alone vs. in complex: kinetics of solvent accessibility were compared for each peptide using ANOVA. Peptides with a Benjamini–Hochberg corrected $P$ value $\leq 0.01$ and 3 successive time points presenting an absolute difference of RDU > 4 times the mean standard deviation between replicates in the entire dataset were considered regulated and colored in the final figures. Otherwise, peptides were colored in gray in the heatmaps and 3D figures of differential analyses. **d** RDUs of the smallest peptides were mapped to the amino-acid portion they cover on the protein sequence. These were then presented as color gradients on heatmaps: light green and dark blue correspond to low and high RDU, respectively. The origin of RDU reported on the heatmap is indicated above the protein sequence with peptides represented as solid and dashed lines when their RDUs were reported or ignored, respectively. Residue numbers are indicated on the top. **e** 3D visualization of the RDU (left panel) or significant differences of RDU (right panel) measured in this study: blue and red correspond to a reduction or an increase of RDU in the complex, respectively. The mapping of peptide-level data to protein sequence was performed as described in **d** with the sum of RDU per timepoint (sum(mean(RDU))) or the sum of the mean of difference of RDU per timepoint (excepted T0) (sum(mean(diff. RDU))). H hydrogen, D deuterium, diff. difference, RDU relative deuterium uptake.

kept only the RDU (or difference of RDU) of the shortest one. Finally, we mapped the RDUs to the protein 3D structures in order to visualize the regions the most accessible to solvent (Fig. 1e, left panel), or presenting significant differences of deuteration (Fig. 1e, right panel). For differential analysis, we colored the 3D structures with the sum of RDU differences between the two conditions (the values per timepoint are available in Supplementary Data 1–3). In this context, modification of solvent accessibility upon complex formation can be due to the presence of a binding interface or to allosteric changes. This approach was employed to identify the conformational differences between the std20S and i20S, and to compare their structural changes upon

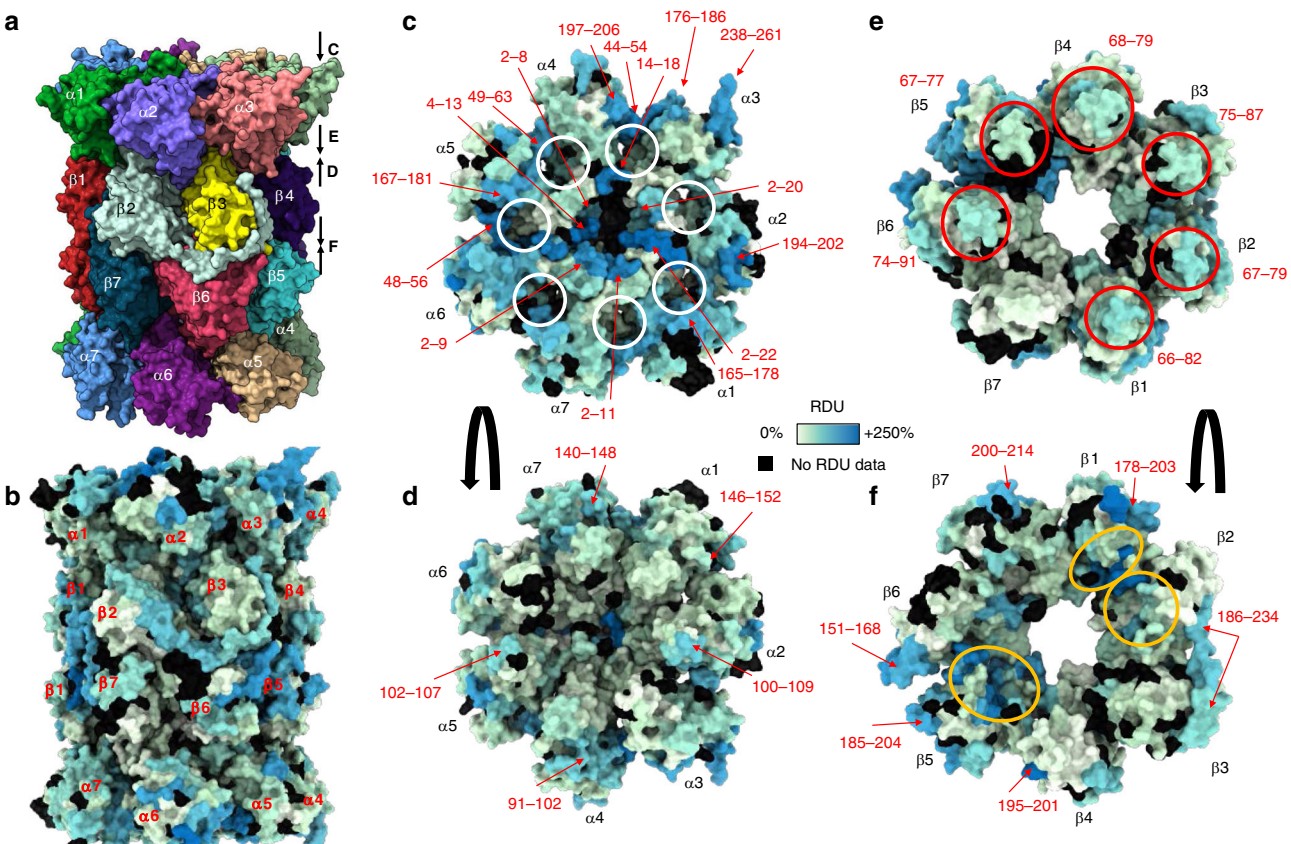

**Fig. 2 HDX-MS of the std20S reveals a dynamic pore entrance and inter-ring interface. a, b** Structure of the human std20S proteasome (PDB: 5LE5)[35] showing the four stacked αββα heptameric rings colored by subunit (**a**) or relative deuterium uptake (RDU) (**b**). **c-f** Their solvent accessibility is represented on each face of the α-ring (solvent interface (**c**) and β-ring interface (**d**)) and β-ring (α-ring interface (**e**) and β-ring interface (**f**)) as the sum of consolidated relative deuterium uptakes between 0.5 and 30 min (described in Fig. 1b, c). The α-pockets (docking sites for the C-termini of PA28 regulators) are circled in white in **c**. Regions discussed in the text are indicated with red arrows and the moderately flexible/accessible bulges at the interface between the α- and β-ring are circled in red in **e**. The corresponding residues are indicated in red around the structures. Active sites are circled in orange (**f**). The ring faces visualized in **d–f** are indicated by arrows on the right of the structure in **a**.

binding to PA28αβ or PA28γ. For this, we purchased commercial i20S purified from human spleen previously used for in vitro proteolytic assays[42–44].

We obtained 89% subunit sequence coverage on average for the 17 subunits of the std/i20S (Supplementary Fig. 1, Supplementary Data 4 and 5, Supplementary Table 1). HDX-MS results were mapped on the 3D structure of the human std20S (PDB: 5LE5, Fig. 2a, b). As expected, we found a faster and higher deuteration of the solvent-facing α-ring interface of the std20S compared to any other ring interfaces (Fig. 2c). This observation benchmarks our approach by confirming that high RDUs directly indicate regions of the proteasome that are more dynamic and potentially accessible for interaction with regulators or PIPs. The N-terminal (N-ter) part of the α-subunits constituting the α-ring were highly deuterated (2–22 in α1, 2–20 in α2, 14–18 in α3, 2–8 in α4, 4–13 in α5, 2–9 in α6, and 2–11 in α7, Fig. 2c). These N-ter are located around the proteasome pore entry, and their high RDU strongly supports the hypothesis of a flexible pore entrance in the std20S that fluctuates between open and closed states[45–47]. Furthermore, the α-ring presented very dynamic patches consisting of non-contiguous peptides from the same subunit in α3 (176–186/ 238–261), α4 (44–54/197–206), and α5 (49–63/167–181) (Fig. 2c). These may interact with many different PIPs. Our analysis also identified dynamic patches at the interface between different subunits like α1 (2–22/165–178) and α2 (194–202), α3 (14–18) and α4 (44–54), and α5 (167–181) and α6 (48–56) that

correspond to the 1–2, 3–4, and 5–6 α-pockets[48] (Fig. 2c, circled in white) known to accommodate the C-terminal (C-ter) tails of Rpt3, Rpt2, and Rpt5, respectively[23,48]. We also identified flexible patches on the interface of the α-ring facing the β-ring, where amino-acid stretches of α1 (146–152), α4 (91–102), α5 (102–107), and α7 (140–148) were slightly more flexible than the remaining of the interface (Fig. 2d and Supplementary Data 3, 6).

The β-ring was globally less deuterated than the α-ring (Fig. 2e, f), which was expected since it does not present a large solvent interface. However, after 30 min, some α-facing bulges were moderately deuterated in all β subunits but β7: these include the loops between α-helices 1 and 2 of β1 (66–82), β2 (67–79), β3 (75–87), β4 (68–79), β5 (67–77), and β6 (74–91) (red circles in Fig. 2e, and Supplementary Data 6). Interestingly, some of these dynamic loops face the highly deuterated regions of the α-ring described above and define flexible regions of the α-ring/β-ring interface. The most dynamic regions of the β-ring were located on its outer surface and were constituted of the C-ter of β1 (178–203), β2 (186–234), β4 (195–201), β5 (185–204), β6 (151–168), and β7 (200–214) (Fig. 2f in red and Supplementary Data 6). These regions are particularly interesting since they can potentially interact with the numerous PIPs described in the literature.

**An inner-to-outer conformational change upon substitution of standard to immuno-subunits.** Although mouse i20S and std20S

possess very similar β2 and β2i substrate binding channels, there are structural differences between their standard and immuno-catalytic subunits β1 and β5. The S1 and S3 substrate pockets are smaller in β1i vs. β1, β5i possesses smaller S2 and S3 pockets, and the S1 pocket is larger in β5i than in β5[31]. However, very little structural difference is found between the α and noncatalytic β subunits of the i20S and std20S (Root-Mean Square Deviation of the Cα < 0.72 Å). The same is true when comparing the recent structure of the human i20S[36] with the std20S[35] (RMSD of the Cα < 0.67 Å).

The std20S catalytic chamber—localized at the β-ring/β-ring interface—was locally very flexible in our data (Fig. 2f and Fig. 3). The residues forming the S1 and S2 pockets[31] of β1 were highly deuterated, together with the T23 of the S3 pocket (Fig. 3b, see Supplementary Data 4 for full kinetics). The catalytic residues of β2 were poorly deuterated compared to the remainder of this subunit (Fig. 3c), with the exception of 45–53 constituting its S1 and S2 pockets. Conversely, most of the residues forming β5 catalytic site and substrate pockets were highly deuterated (Fig. 3d). The relatively poor sequence coverage that we obtained for β2i limits its analysis and we cannot comment on the accessibility of its S1, S2, and S3 pockets (Fig. 3b and Supplementary Table 1). The direct comparison of RDUs between the standard and immuno-catalytic subunits (β1/β1i, β2/β2i, and β5/β5i) was not possible due to their sequence differences. Nevertheless, we could acquire the deuteration profiles of the peptides ADSRATAGAY (16–25) of β5 and AVDSRASAGSY (15–25) of β5i that cover the same portion of the two subunits. Their difference of deuteration profiles indicate that these residues are more flexible in std20S than in the i20S (Fig. 3d and Supplementary Data 4).

Allosteric differences beyond the catalytic subunits were shown in the prokaryotic std and i20S analysed by NMR[34]. These were not observed in mouse crystallographic structures[31], may be due to crystal packing. In order to confirm or infirm this inner-to-outer allosteric change in human, we compared the RDUs of all the noncatalytic subunits of the i20S (Supplementary Data 4–6) with those of the std20S (as presented Fig. 1).

The introduction of immuno-catalytic subunits resulted in significant conformational changes on the noncatalytic 20S subunits. The results of our statistical analysis are presented in Fig. 4, and Supplementary Data 1, 4, 7–10. Overall, the noncatalytic β-ring subunits were more dynamic in the i20S than in the std20S, (>50 peptides significantly more deuterated, Fig. 4a–b in red, Supplementary Data 1, 7). The β-ring/β-ring interface and the channel were particularly more dynamic/accessible in the i20S vs. std20S (Fig. 4a/e and Supplementary Data 1), together with two α-facing bulges of β6 and β7 (Fig. 4b, red circles). Four β-facing bulges of α2/3/5/6 were also significantly more dynamic in the i20S vs. std20S (Fig. 4c, red circles, corrected P-values and RDU differences are available in Supplementary Data 9 and 10). The pore entrance on the α-ring solvent-accessible surface (α subunits N-ter) was significantly more dynamic/accessible upon replacement of β1/2/5 by the immuno-subunits (Fig. 4d red circles, Supplementary Data 1, 7). This suggests a long-range mechanism enabling the i20S pore to dwell longer in the open-state, and could explain its generally higher activity[49] as well as the various regions from the antechambers that were more accessible in the i20S vs. std20S (98–99 in α2; 102–109 in α3; 128–142 in α5; 100–108 in α6; 95–105 in β3; 94–102 in β4, Fig. 4e, corrected P-values and RDU differences are available in Supplementary Data 9 and 10). Conversely, three solvent-facing regions of the α-ring were significantly more dynamic/accessible in the std20S than in the i20S (Fig. 4d in blue, Supplementary Data 4, 7). Since these

constitute potential binding sites for the proteasome regulators, this could explain the preferential association of specific regulators to different 20S subtypes. Interestingly, these three subunits also contain regions that are more accessible in the std20S on the β-ring facing interface (Fig. 4c), suggesting that this dynamic behavior can be related to the presence of the standard catalytic subunits within the β-ring. More precisely, the α1 102–114 (corrected P value = 5.87.10$^{-11}$, sum of difference = −12%) and 146–152 (corrected P value = 2.54.10$^{-7}$, sum of difference = −12%), which were more deuterated in the std20S vs. i20S, are in direct contact with a portion of β2/β2i (69–72) also more accessible in the std20S (Supplementary Data 1). Altogether, these results confirm the inner-to-outer allosteric effect following incorporation of the immuno-β subunits that lead to α-ring remodeling: more dynamic pore entrance but protected external anchor regions (α3 C-ter).

**The activation loop of PA28β is less dynamic/accessible than PA28α/γ's.** There is no structure yet for PA28γ but those of the human homoheptameric PA28α (nonphysiological) and murine heteroheptameric PA28α$_4$β$_3$ were resolved in 1997[32] and 2017[33], respectively. These monomers are composed of a four-helix bundle, assembled as a barrel-shaped heptamer (Fig. 5) that controls proteasome catalytic activity through a conserved activation loop[50,51].

The structures of PA28α, PA28β, and PA28αβ miss a region of 15–31 residues bridging the α-helices 1 and 2, although they were present in the recombinant constructs (Fig. 5a–c and Supplementary Data 11 in red). These apical loops, most likely flexible, sit on top of the regulator and might thereby control the substrate access to the proteasome. The human PA28γ chain has a similar loop (43 residues), and no information on its structure has ever been reported. Since the sequence identity of human PA28γ is closer to PA28α (40%) than PA28β (35%) or *Plasmodium falciparum* (Pf) PA28 (34%) (Supplementary Table 2), we generated a homology model of human PA28γ (Fig. 5e) using the structure of human PA28α[33] as a template with the SWISS-MODEL server[52]. The sequence of Pf is used as a comparison here because it is part of the very few structures of PA28 regulator homologues available in the PDB with close homology to PA28γ and whose structure was recently solved in complex with the 20S[24].

The sequence coverages obtained upon digestion were above 90% for all three PA28 subunits across all conditions (Supplementary Fig. 1, Supplementary Data 11, 12 and Supplementary Table 1). Their deuteration pattern revealed similar dynamics for both PA28αβ and PA28γ with a very strong protection of most residues inside the channel (Fig. 5d, e and Supplementary Data 11). The missing apical loops (Supplementary Data 11, red rectangles) were very quickly deuterated, a sign of high accessibility/flexibility, that could explain their absence in known structures. The proteasome-facing interface (unstructured N- and C-termini and activation loops) (Fig. 5d, e, right) and the entrance of the channel (Fig. 5d, e, left) were also strongly deuterated. Remarkably, the regions involved in the binding (C-ter) and activation of the proteasome were very accessible/dynamic when both regulators were alone in solution. However, the activation loop was less deuterated in PA28β. The reason why eukaryoric homoheptameric PA28 regulators have evolved towards more complex heteroheptameric PA28αβ is not well understood. Given the overall similar tridimensional structures of PA28αβ and PA28γ, the poorer flexibility observed on PA28β activation loop could, at least partially, explain their different functions.

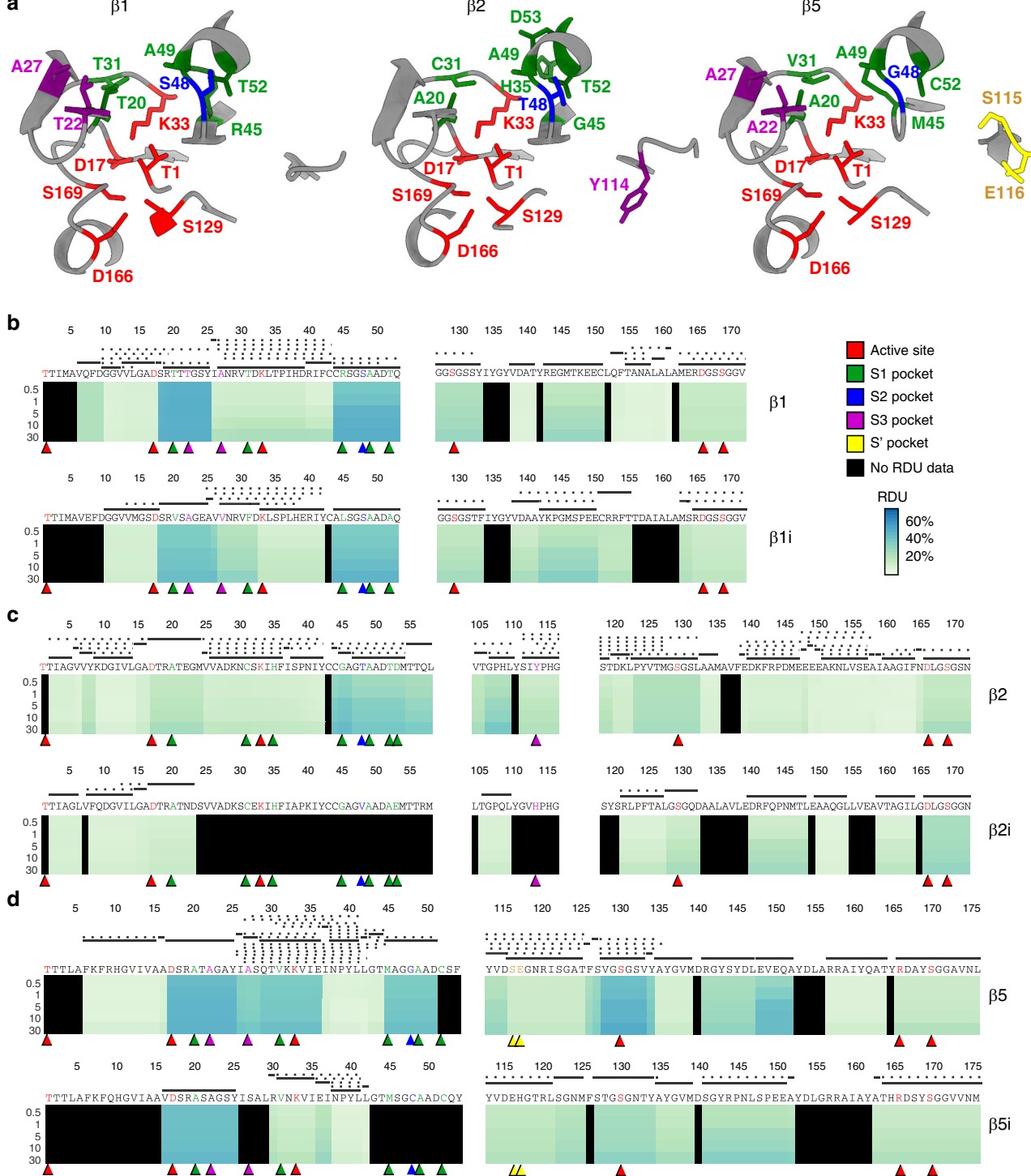

**Fig. 3 Comparison of the catalytic-subunit solvent accessibilities. a** Representation of the three catalytic subunits β1 (left), β2 (middle), and β5 (right) of the std20S (PDB: 5LE5[35]). The residues forming the active site and the substrate-specific pockets are indicated and color-coded (active site, S1, S2, S3, and S' are represented in red, green, blue, purple, and yellow, respectively). **b–d** Deuteration heatmaps showing the relative deuterium uptake (RDU) for each timepoint of the kinetics, along the protein sequences of β1/β1i (**b**), β2/β2i (**c**), and β5/β5i (**d**). The peptide sequence coverage is presented above the heatmaps as explained in Fig. 1d. For the sake of clarity, only the sequence stretches containing residues of the active sites and substrate pockets are represented here.

## Allosteric changes between PA28γ and PA28αβ upon 20S binding

Before comparing the PA28 regulators alone and in complex with the std20S, we confirmed their ability to increase the proteasome activity in vitro. In our hands, a 2-fold molar excess of any of the PA28 regulators doubled the std20S chymotrypsin-like activity (P value of a paired two-sided t-test: 0.0007 and 0.0013 for PA28γ and PA28αβ, respectively; Supplementary Fig. 2 and Supplementary Table 3). As expected, the PA28 proteasome-facing interfaces were highly protected in the complex compared to the regulators alone (Fig. 6a, b right, in blue; Supplementary Data 2, 12,

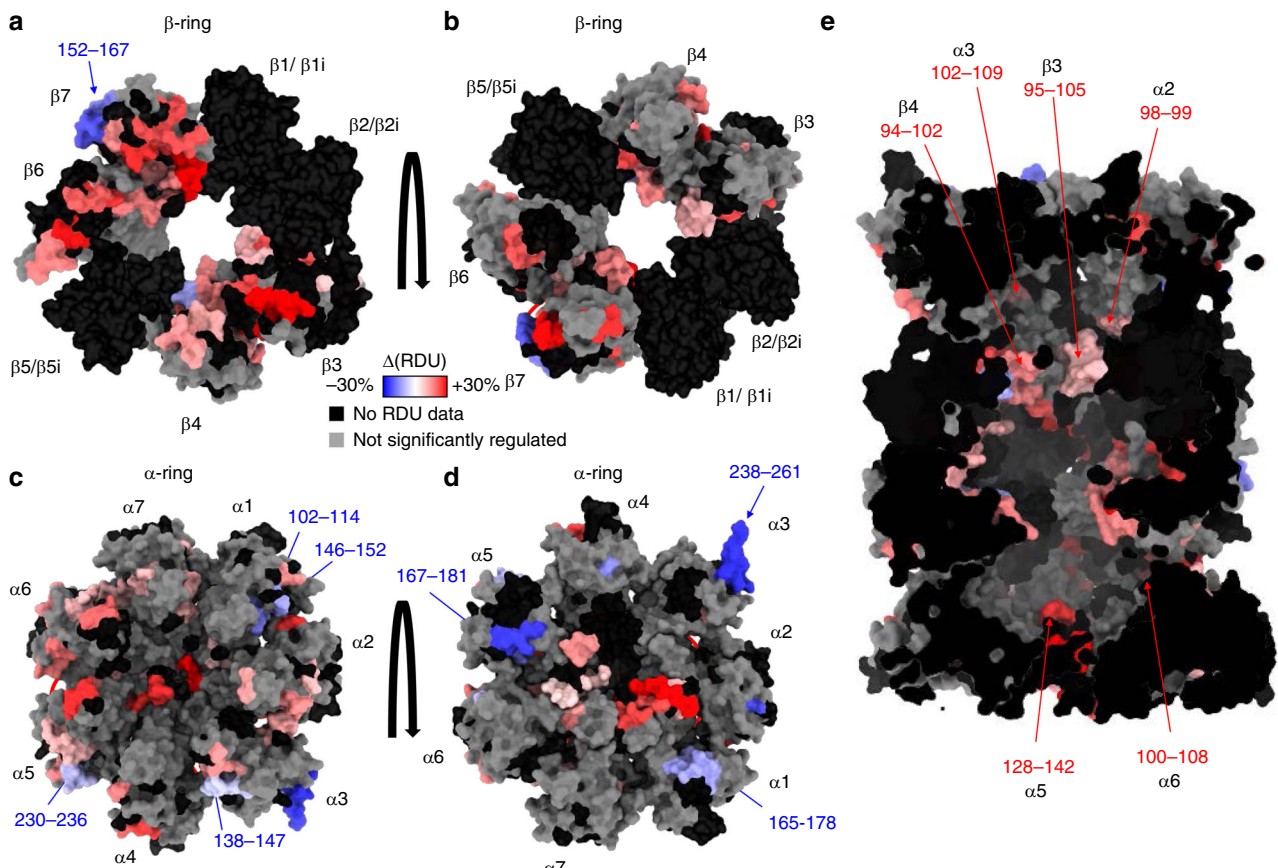

**Fig. 4 Differential analysis of std20S and i20S. a–d** Sum of differential relative uptakes between the i20S and the std20S ($\Delta(\text{RDU}) = \text{RDU}_{i20S} - \text{RDU}_{std20S}$): β-ring/β-ring interface (**a**); β-ring facing the α-ring (**b**); α-ring facing the β-ring (**c**); solvent-facing α-ring (**d**); inside view showing the antechambers and catalytic pockets (**e**). Color-coded regions were significantly more accessible/dynamic in the std20S (blue = −30%) or i20S (red = +30%) and regions of interest are indicated with arrows. Regions that are not covered in one or both conditions, or do not pass the statistical threshold are represented in black and grey, respectively (see Fig. 1 for more details). RDU relative deuterium uptake.

and 13). The activation loops, located at the 20S binding interface, and known to interact with the N-ter of the 20S α subunits to open its central pore, were highly dynamic/accessible in both PA28 analyzed alone (Fig. 5d, e right, and Supplementary Data 11 and 12). Our data show that PA28αβ/std20S binding strongly protected the activation loops as well as their neighboring residues (136–147, 153–159, and 232–249 in PA28α; 123–135, 140–145 in PA28β, Fig. 6a and Supplementary Data 2, 13, corrected *P* values and RDU differences are available in Supplementary Data 9–10). The protection of PA28γ activation loop (region 146–152) upon std20S binding was also statistically significant but notably less pronounced (corrected *P* value = 1.49.10$^{-8}$, sum of difference = −20%) (Fig. 6b and Supplementary Data 9, 10).

Interestingly, other regions were less dynamic/accessible upon complex formation. The kinks of the first α-helix of PA28α (25–32; corrected *P* value = 2.93.10$^{-15}$, sum of difference = −61%) and PA28γ (35–40; corrected *P* value = 6.55.10$^{-13}$, sum of difference = −30%) (Fig. 5a and Fig. 6a, b, red ovals) were strongly protected upon std20S binding. The PA28β N-ter (5–16) that faces PA28α's kink, was also significantly protected upon std20S interaction (corrected *P* value = 9.19.10$^{-8}$, sum of difference = −30%) (Supplementary Data 2). We thus hypothesize that PA28α/PA28β interaction might be strengthened and/or structurally rearranged upon binding to the std20S. The N-ter (6–10) of PA28γ facing the kink was slightly but not significantly protected (corrected *P* value = 3.77.10$^{-3}$, sum of difference = −6.4%) (Fig. 6b and Supplementary Data 2).

The opposite side of the heptamer was also partially protected upon std20S binding: the loops between helices 3 and 4 of PA28α (179–197), PA28β (166–184), and PA28γ (203–208) (Fig. 6a, b) as well as the flexible loop (60–101) of PA28γ (Fig. 5b). These regions include the constriction sites at the top of PA28αβ (K187 and K190 in PA28α, K177 and K180 in PA28β) that were significantly protected upon std20S binding (Fig. 6a and Supplementary Data 2). In PA28γ, another constriction site, located underneath was protected upon std20S binding (R181, Fig. 6b and Supplementary Data 2).

We then compared the impact of std20S vs. i20S binding on PA28 solvent accessibility using the statistical strategy presented in Fig. 1c (comparison referred to as PA28 + std20S vs. PA28 + i20S in Supplementary Data 2, 9, 10). Most of the reduction in accessibility/flexibility that was observed on the PA28 chains upon binding to the std20S were dampened upon interaction with the i20S (Fig. 6c, d and Supplementary Data 2).

**A PA28-driven outer-to-inner allosteric change of the 20S proteasome.** Atomic force spectroscopy showed that the 20S can alternate between open and closed states, possibly through allosteric regulation[45,46]. Our data indicate that PA28αβ and PA28γ do not present the same conformational rearrangements upon binding to the 20S, so they might allosterically affect the 20S pore entrance and/or its catalytic sites in a regulator-specific outer-to-inner mechanism, as suggested recently[25].

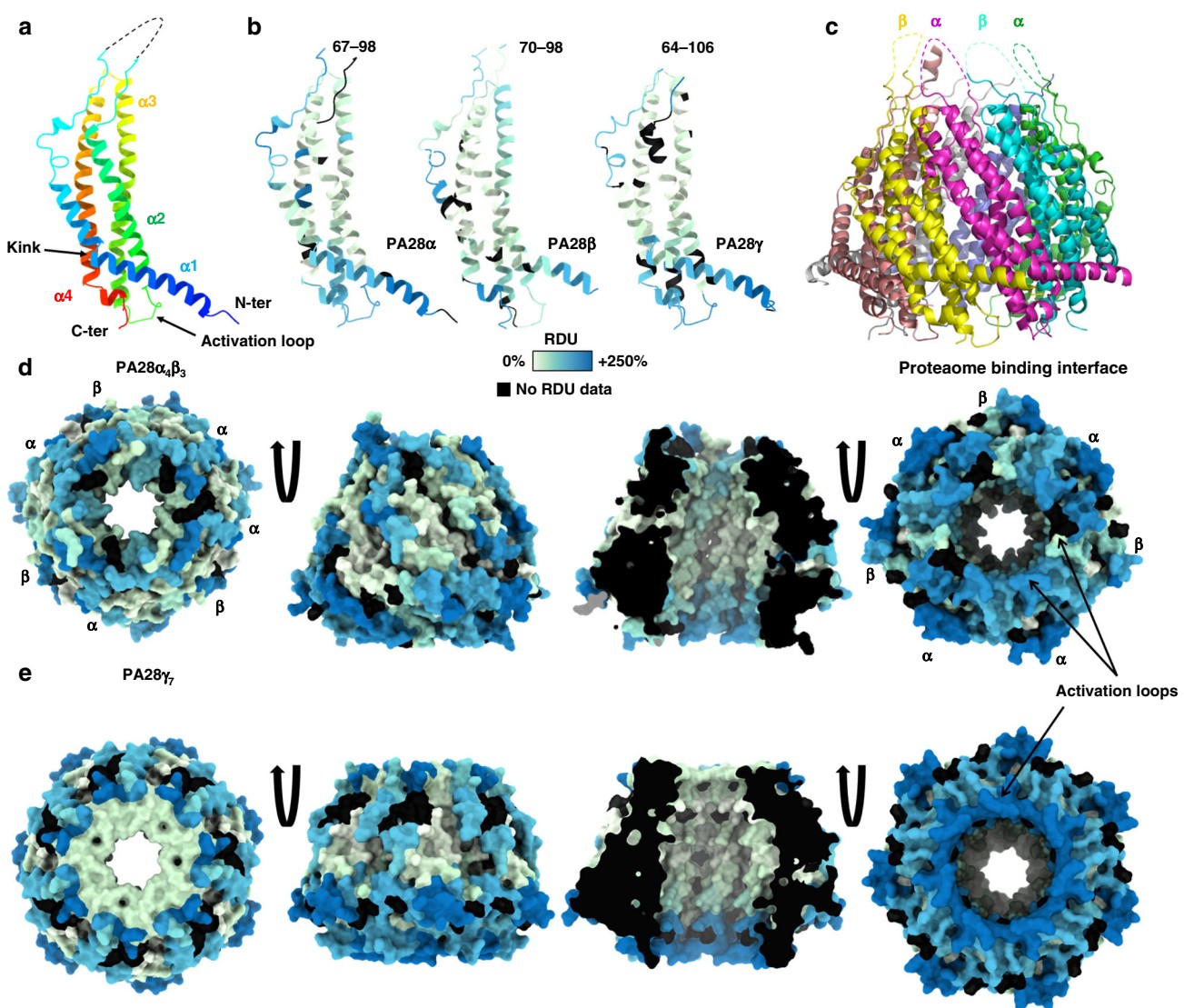

**Fig. 5 PA28 regulators present flexible loops on both ends of the channel. a, b** Structure of monomeric PA28α, β, or γ showing the 4-helix-bundle colored based on amino-acid numbering (**a**) or the sum of hydrogen-deuterium relative uptake accumulated over 30 min in PA28γ or PA28α$_4$β$_3$ in solution (**b**). The residues corresponding to the missing apical loops (dashed lines) are indicated in **b. c** Structure of the heptameric complex PA28α$_4$β$_3$ (PDB: 5MX5[33]). Only four apical loops out of seven are represented, for the sake of clarity. **d, e** Surface accessibility of the heptameric complexes PA28α$_4$β$_3$ (**d** PDB: 5MX5) and PA28γ$_7$ (**e** homology model) viewed from the top (left), front and inside (middle), and bottom (right). Regions that were not detected are represented in black (described in Fig. 1b, c). RDU relative deuterium uptake.

We compared the std/i20S RDUs in presence or absence of PA28αβ or PA28γ (Fig. 7, Supplementary Data 3, 14, and 15). Strikingly, very few regions of the α-ring in contact with the regulators were significantly protected in the std20S upon interaction with PA28αβ or PA28γ, and none in the i20S. The same C-ter region of α3 (238–261) that was found to be more accessible in the std20S vs. i20S (corrected $P$ value = 1.55.10$^{-7}$, sum of difference = −24%) (Fig. 4d) was protected in the std20S upon interaction with PA28αβ (corrected $P$ value = 1.62.10$^{-5}$, sum of difference = −28%), and to a lesser extent with PA28γ (corrected $P$ value = 2.85.10$^{-4}$, sum of difference = −16%) (Fig. 7a, f, red circle on α3). Other regions of the std20S that were protected when binding PA28αβ form two patches on the same solvent-exposed interface (Fig. 7a, red circles on α4, α5). These could be the main anchorage sites of PA28αβ at the surface of the std20S. The remainder of the std20S solvent-exposed surface was globally more dynamic upon PA28αβ binding, especially near the pore entrance (N-ter of α1/2/3/4/5/7), which can be interpreted as an opening of

the pore (Fig. 7a, large red circle). Another external patch at the α1/α2 interface was also more dynamic upon binding of PA28αβ (Fig. 7a, red circle on α1, α2).

On the β-facing α-ring, PA28αβ binding mainly destabilized α2 (136–145: corrected $P$ value = 2.48.10$^{-5}$, sum of difference = 10%) and the α5 bulge (102–107: corrected $P$ value = 1.56.10$^{-5}$, sum of difference = 19%) (Fig. 7b, red circles on α2, α5). Interestingly, this bulge faces the 109–135 β6 stretch encompassing 4 peptides more deuterated upon PA28αβ binding and is in close vicinity to the β5 interface, which is also more dynamic (Fig. 7c, red circles on β5, β6, corrected $P$ values and RDU differences are available in Supplementary Data 9–10). Although not close enough to make hydrogen bonds, α2 (136–145) (Fig. 7b, red circle on α2) faces the β2/β3 interface that was more flexible/accessible upon PA28αβ binding, especially in the 186–202 C-ter region of β2 extending towards β3 (Fig. 7d, circle on β2). The only peptides of the β-ring that were less dynamic upon PA28αβ binding are located on β1 (Fig. 7c, red circle on β1).

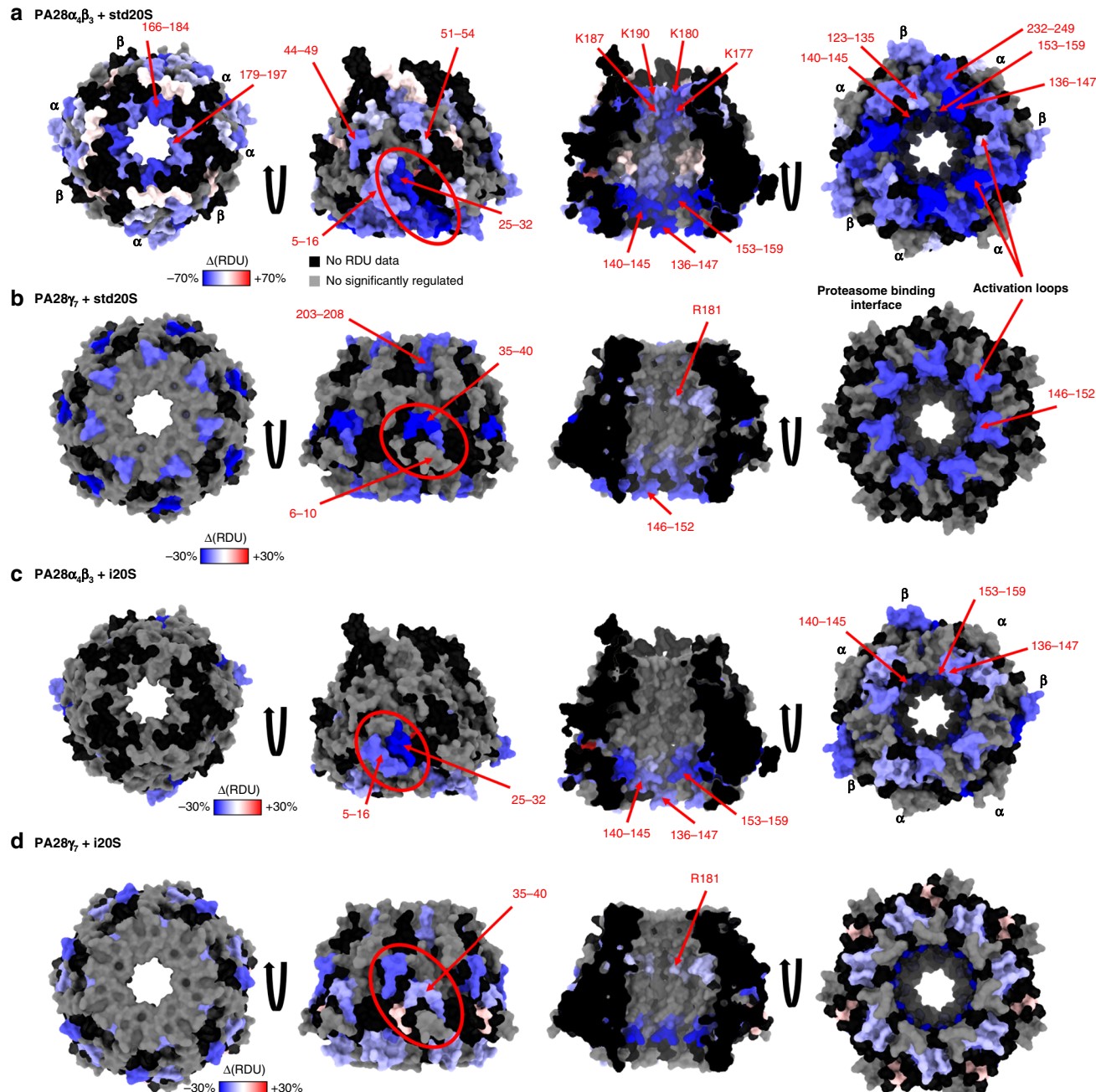

**Fig. 6 Activation of PA28αβ and PA28γ by the 20S involves a rearrangement of the N-ter kink and an allosteric change.** The sum of differential RDUs accumulated from 0.5 to 30 min deuteration of the PA28 regulators alone or in presence of the 20S proteasome (Δ(RDU) = RDU$_{PA28/20S}$ − RDU$_{PA28}$) is color coded from blue (less accessible in the complex) to red (more accessible in the complex) and ranges from −70%/+70% for PA28αβ + std20S (**a**) and −30%/+30% for PA28γ + std20S (**b**) PA28αβ + i20S (**c**) and PA28γ + i20S (**d**). The heptameric complexes PA28α₄β₃ (**a** and **c** PDB: 5MX5) and PA28γ₇ (**b** and **d** homology model) are viewed from the top (left), front and inside (middle), and bottom (right). Residues of PA28 that present a significant decrease of deuteration upon binding of the 20S proteasome are located around the N-ter kink (red ovals), the activation loop and 20S interface and the entry of the channel (left). Residues in gray were not considered significantly different according to the statistical thresholds presented in Fig. 1c. Residues with no RDU information in one or both conditions are in black. Δ(RDU) sum of differences of relative deuterium uptake.

We compared the effect of PA28αβ binding to the std20S (Fig. 7a–e) with the other conditions (Fig. 7f–t). Although the destabilization of the α-ring N-ter by PA28αβ was almost complete in the std20S (except α6) (Fig. 7a, large red circle), it was restricted to α2/3/7 in the i20S (Fig. 7k, large red circle), suggesting a partial opening of the pore. Binding of PA28γ did not affect any of the std20S α subunit N-ter (Fig. 7f) and only the i20S α2 and α7 N-ter (Fig. 7p, large red circle), indicating less PA28γ-driven opening of the std20S than the i20S pore.

Overall, PA28 binding to the std/i20S increased their RDU, especially at the pore entrance, at the α1/α2 interface (Fig. 7a, k, p, red circles on α1, α2), and the β2/β2i C-ter arm extending towards β3 (Fig. 7d, e, n, o, s, t). Despite these similarities, our data show subtle proteasome- as well as regulator-specific - differences in the allosteric motion triggered upon complex formation, as suggested recently[25]. The std20S α3 C-ter was protected upon binding of PA28αβ and PA28γ (Fig. 7a, f). This seemed to induce the PA28γ-specific destabilization of α2

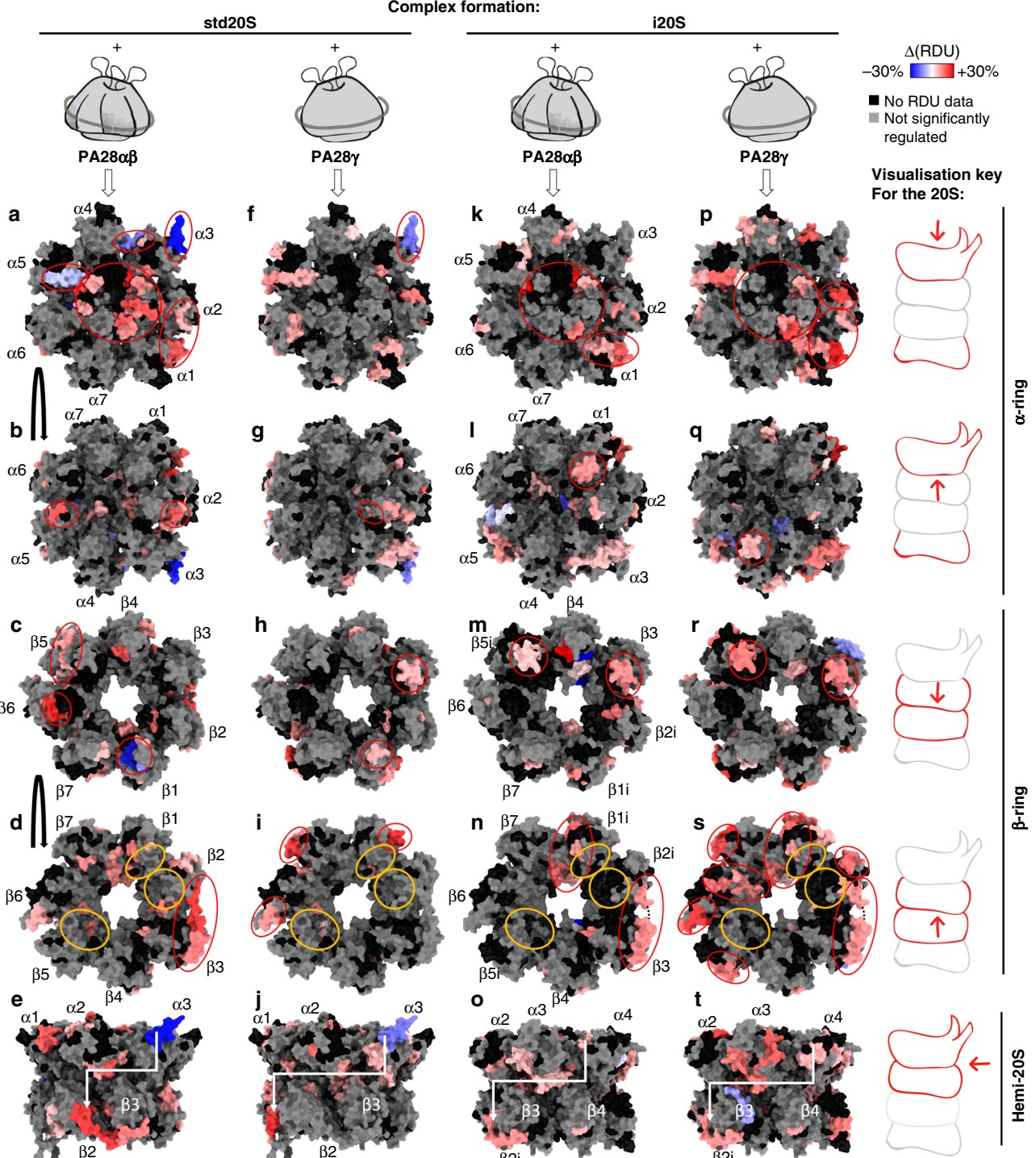

**Fig. 7 Catalytic-subunit- and regulator-specificity of the outer to inner allosteric changes.** Differential relative uptakes accumulated over 30 min deuteration of the 20S proteasomes with and without the PA28 regulators ($\Delta$(RDU) = RDU$_{20S/PA28}$ − RDU$_{20S}$). The regions that were statistically protected or more dynamic upon regulator binding are represented in blue and red, respectively, for the std20S (PDB: 5LE5) and PA28αβ (PDB: 5MX5) (**a**–**e**), the std20S and PA28γ (**f**–**j**), the i20S (PDB: 6E5B) and PA28αβ (**k**–**o**), and the i20S and PA28γ (**p**–**t**). Active sites at the β-ring/β-ring interface are circled in orange. The side views of the hemi-proteasome **j/o/t/e** show the α-ring and β-ring interface. Residues in gray were not considered significantly different according to the statistical thresholds presented in Fig. 1c. Residues with no RDU information in one or both conditions are in black. $\Delta$(RDU) sum of differences of relative deuterium uptake. Regions discussed in the text are circled in red. Missing residues from the C-terminal part of β2i are represented as a black dotted line in panels **n** and **s**. The visualization key on the right indicates the identity and orientation of the structures with the same schematics of the 20S as in Fig. 8.

(101–119: corrected $P$ value = 4.03.10$^{-5}$, sum of difference = 13%, Fig. 7g, red circle on α2), β3 (74–86: corrected $P$ value = 1.19.10$^{-6}$, sum of difference = 14%), and β1 (64–70: corrected $P$ value = 8.72.10$^{-4}$, sum of difference = 10%) bulges (Fig. 7h, red circles on β1, β3), down to the outer β-ring/β-ring interface of β1 (178–205), β6 (151–159: corrected $P$ value = 1.40.10$^{-7}$, sum of difference = 13%) and β7 (152–167) (Fig. 7i, red circles on β1, β6, β7; see Supplementary Data 8 for the corrected $P$ values and RDU differences corresponding to the protein portions covered with multiple peptides). According to our statistical thresholds, the α3 C-ter protection upon PA28 binding was std20S-specific (Fig. 7k, p), but the remodeling of the α-ring occurred in both 20S (although differently) and was propagated to the β-ring of the i20S via α1/β3/β5i and α4/β3/β5i bulges for PA28αβ (Fig. 7l, m, red circles on α1/β3/β5i) and PA28γ (Fig. 7q, r, red circles on α4/β3/β5i). It resulted in remodeling of the β-ring/β-ring interface eventually affecting β1i/2i/7 or β1i/2i/5i/6/7 with PA28αβ (Fig. 7n, red circles on β1i/2i/7) and PA28γ (Fig. 7s, red circles on β1i/2i/5i/6/7), respectively.

The limited sequence coverage around the catalytic sites prevented us from drawing exhaustive conclusions on the differential impact of regulator/20S complex formation on the proteasome active sites. However, β1 (10–25) (corrected $P$ value = 9.49.10$^{-9}$, sum of difference = 22%), β2 (16–42) (corrected $P$ value = 9.54.10$^{-5}$, sum of difference = 5.2%) and β5 (29–36) (corrected $P$ value = 8.84.10$^{-3}$, sum of difference = 8%) active sites were partially but significantly more dynamic/accessible upon PA28αβ binding (Fig. 7d, orange circles and Supplementary Data 3 and 14), in-line with an increased proteolytic activity. Binding of PA28αβ also increased the solvent accessibility of β1i active site (10–17) (corrected $P$ value = 3.89.10$^{-3}$, sum of difference = 14%) (Supplementary Data 3 and 14), which is in good agreement with previous work showing that the caspase-like activity of the i20S is enhanced to a higher extent by PA28αβ than PA28γ[53]. PA28γ was reported to have little or no effect on the i20S[44], whereas it is known to increase all three std20S activities[54]. Our data do not indicate any major change in the std20S active site solvent accessibility upon interaction with the regulator (Fig. 7i). However, it revealed a significant decrease of β2i N-ter (2–6) (corrected $P$ value = 3.80.10$^{-3}$, sum of difference = −13%) solvent accessibility upon PA28γ interaction (Supplementary Data 3). Thus, the catalytic activation observed upon PA28γ/std20S binding may be driven by changes occurring further away from the active sites.

## Discussion

This structural dataset on the human i20S compared to the std20S alone or bound to both PA28 regulators rationalizes from a mechanistic point of view previous observations that replacement[49], modification[34] or ligand binding[49,55] to the catalytic subunits can allosterically modify the 20S core particle structure and alter its binding to potential PIPs. A main advantage of HDX-MS is that it provides information on very flexible loops, unlike X-ray crystallography or cryo-EM. Also, it is not theoretically limited by the complex size since the proteins are digested into peptides after deuteration: it can be applied to very small proteins (unlike cryo-EM) or very large complexes (unlike NMR). Large oligomeric complexes containing a reduced number of different monomers (limited number of chemical shifts) have already been analyzed by NMR, such as the prokaryotic 20S from *T. acidophilum* constituted of 14 identical α and 14 identical β subunits[34]. With the recent exception of the DNA-PKc analysis[56] (469 kDa monomer), HDX-MS is usually performed either on rather small and simple systems[39] or on homo-oligomeric complexes[38,57], due to some technical limitations in protein digestion, peptide separation and data analysis. In comparison,

our study encompasses 16 different ~25 kDa monomers. Despite this analytical challenge, we obtained an average sequence coverage per subunit of 89% for the 20S alone and 81% when in complex with PA28 regulators. It is important to note that, except for a few specific cases[58,59], HDX-MS does not inform on the presence of multiple conformational states in solution since it is based on the measured average of their RDUs. Other approaches such as cryo-EM-based classification would be needed to better characterise intermediate or minor states. Nevertheless, comparing the same sample (like std20S) in presence or absence of regulator and performing thorough statistical analysis identified a wealth of information on the proteasome dynamics, binding interfaces and allosteric changes upon incorporation of the immuno-subunits or activation by PA28 regulators (Fig. 8).

These conformational maps revealed the subtle rearrangement of the std/i20S α3 C-ter. More precisely, the residues 238–261 were less dynamic/accessible upon incorporation of the immuno-catalytic subunits (in blue Fig. 8a). The principal component analysis of the 20S high-resolution structures available in the Protein Data Bank (PDB) in 2014 (46 structures from yeast, 4 murine and 1 bovine) revealed that the 220–230 stretch of the mouse std20S and i20S α3 clustered with the yeast apo and peptide-bound forms, respectively[49]. We believe that the allosteric change occurring on α3 C-ter upon ligand binding to β5 in the yeast 20S could be similar to the change in solvent accessibility/dynamics observed between the std20S and the i20S conformational maps. The highly charged C-ter of α3 points towards the outer surface of the α-ring and has also been suggested to be the Insulin-Degrading-Enzyme binding site[60]. It could be involved in the binding of PA28αβ/γ to the std20S, or in its subsequent activation. Interestingly, the very end of α3 C-ter is absent from most human 20S proteasome structures, which confirms its high flexibility but also impedes any modeling of 20S/PA28 interaction. Unfortunately, this protruding C-ter α helix is not present in the cryo-EM structure of the *Pf* 20S bound to its PA28 regulator, since it is nine residues shorter than its human counterpart[36]. It could be characteristic of more evolved 20S proteasomes and tune their binding with different regulators.

As commented before, the differential analysis of the catalytic subunits is impossible due to their sequence differences (the peptides resulting from the pepsin digestion are different). The comparison of their deuteration profiles was further hindered by the limited sequence coverage of the i20S. This was partly due to the presence of residual traces of std20S present in this sample[61]. Nevertheless, it was possible to compare the RDUs of the non-catalytic subunits of the two core particles. The central pore of the 20S was more flexible in the i20S vs. std20S (Fig. 4d and in red Fig. 8a), as was the β-facing interface of the α-ring, especially α5 that directly faces the β5 (std20S) or β5i (i20S) catalytic subunit (Fig. 4c). We propose that the conformational differences observed between the standard and immuno-α3 C-ter (red arrows Fig. 8a) could be transduced from the inner β-ring to the outer α-ring via α5.

Interestingly, upon binding of any regulator to the 20S, we observed a similar destabilization of the α5 region that faces β5/5i, suggesting that the exact opposite mechanism (from the outer α-ring to the inner β-ring) can also take place upon 20S/PA28 interaction, as suggested earlier[34].

Overall, our data indicate a general mechanism of PA28 activation by the 20S core proteasome. More precisely, the interaction of PA28 activation loops and C-ter with the 20S seems to trigger a long-range allosteric change at the top of the regulator (loops between helices 1/2 and/or 3/4) via a rearrangement of the N-ter kink of PA28α and PA28γ (Fig. 8b). Although present in the three PA28 subtypes, the activation loop protection was stronger in PA28α and PA28γ vs. PA28β; most probably

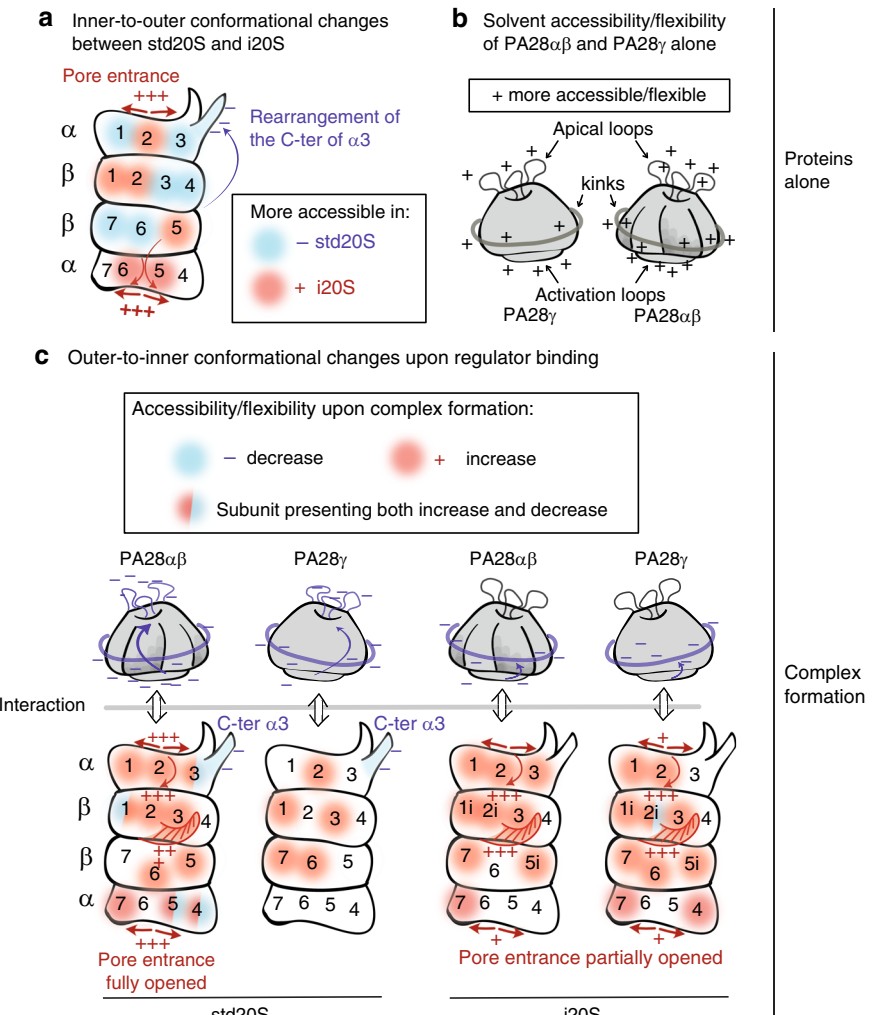

**Fig. 8 Schematic representation of our major finding. a** Comparison of std20S vs. i20S deuteration profiles confirms an inner-to-outer allosteric change: the differences between the std20S- and i20S-specific subunits β1/1i, β2/2i and β5/5i located inside the complex trigger conformational changes at the pore entrance (aperture in the i20S: +++) and external surface of the proteasome, including the C-ter of α3 (inner-to-outer change represented as an arrow). The subunits that are more accessible/flexible in the std20S and the i20S are presented in blue and red, respectively. **b** PA28 solvent accessibilities analysed alone indicate similarities in their dynamic regions: their apical loops, kinks and activation loops were more deuterated (indicated with+). **c** 20S- and PA28-specific reorganizations occurring upon complex formation. This schematics presents the results of the comparison of HDX-MS data from the PA28 and std/i20S alone vs. in complex. The subunits in red and blue were globally more and less deuterated in the complex than in the protein alone, respectively. More localised changes are indicated by half-circles. Opening of the 20S pore entrance is indicated with + and +++ (for minor and more pronounced increase of deuteration) as well as the reorganisation of the C-ter of β2 and β2i upon complex formation. − indicates the area of the proteins that were protected upon binding (apical loops, kinks and activation loops of the regulators; C-ter of α3 in the std20S).

explaining why "the affinity of the proteasome towards the PA28αβ complex is about two orders of magnitude higher than towards the homomeric PA28α and PA28β complexes"[62]. Indeed, several studies showed that both PA28α and PA28β were required for an optimum activation of the 20S by PA28αβ[32,33].

We also provide crucial information on the loops located between helices 1/2, missing in every PA28 high-resolution structure or model. Since these are located at the entrance/exit of the activator, any change in their flexibility could modify the nature of both substrates and/or products of the proteasome complex. Indeed, it has been shown in PA28α to constrain substrates within the catalytic chamber of the core particle, thereby generating shorter peptides[63]. We show that these flexible loops, together with PA28 constriction sites (conserved lysine between alpha helices 3 and 4) are allosterically stabilized upon binding of the std20S but not the i20S (Fig. 8c). This could explain how the binding of different PA28 to different 20S subtypes alters the

length/nature of the generated peptides. Our data also suggest that PA28 might modify the 20S products by inducing conformational changes in the proteasomal active sites. Indeed, our results show distinctive changes in dynamics detected down to the β-ring and around the catalytic site and/or substrate pockets upon regulator binding (Fig. 8c).

We observed a stronger protection of PA28αβ than PA28γ upon std20S binding (Fig. 6 and Fig. 8c). Globally, both PA28 were more protected upon interaction with the std20S than with the i20S (Fig. 8c and Supplementary Data 2 and 13), suggesting a stronger interaction of the regulators with the std20S than the i20S (Supplementary Data 2). This was mirrored by regions on the i20S α-ring that were less protected than in the std20S upon incubation with both PA28, whereas previous studies showed that PA28αβ interaction was stronger with the i20S than with the std20S[64] or equivalent[21]. In this context, we think that protection to deuteration cannot be always correlated to relative binding

affinities. The very limited decrease of RDU on the α-ring interface interacting with PA28 regulators can be explained by the resulting gate opening (RDU increase) known to induce its increased proteolytic activity. Gate opening and binding interface protection are expected to have opposite impacts on RDU, so their combination in the same protein region would result in no significant changes of deuteration rates. We showed that both PA28 regulators activate the chymotrypsin-like activity of the std20S to a similar extent in our in vitro assay (Supplementary Fig. 2). However, the N-ter portions of the std20S α subunits (entrance of the channel) were significantly more deuterated when interacting with PA28αβ whereas this increase was not significant with PA28γ (Fig. 7a, f and Supplementary Data 3), suggesting that their allosteric mechanisms of activation are different. This could alternatively be explained by a shorter dwell-time at the surface of the std20S or by differences in binding stoichiometry. More experiments are needed to fully understand these allosteric mechanisms of activation.

Although not similarly distributed, a global core proteasome destabilization upon complex formation was found in all four conditions tested (Fig. 7), so we focused on the comparison of the most affected regions. We observed that each pair of PA28/20S complex underwent long-range conformational rearrangements in specific regions, as suggested recently[25]. Different sets of α N-ter were destabilized depending on the PA28/20S pair (in red Fig. 8c), suggesting a gradual interaction-specific opening of the central pore, in-line with a recent cryo-EM structure that showed a partial opening of the bovine i20S gate in complex with human PA28αβ[65]. The β2 C-ter extending towards β3 was destabilized in all the conditions, except std20S/PA28γ. On the contrary, the region just before the α3 C-ter (217–236) was protected upon PA28 binding in all cases, except std20S/PA28αβ. Interestingly, this stretch is close to β3 74–86, which was also protected in all conditions tested except std20S/PA28αβ. These observations could provide a basis to understand how regulator binding propagates from the α- to the β-ring depending on the activator/20S pair.

Altogether, these data provide a unique resource informing on the allosteric activation of the 20S proteasome complexes that should soon be completed by upcoming structural and functional studies. We think that HDX-MS could be applied to characterize, not only the binding sites of many new proteasome inhibitors, but also to identify their potential allosteric effects on the 20S core complex. The same is true for the many PIPs that regulate proteasome function and are still poorly characterized from a structural point of view[66]. More broadly, this work demonstrates the ability of HDX-MS to investigate dynamic events on megadalton assemblies including ribosomal particles, inflammasomes, and nucleosomes, to name a few.

## Methods

**Reagents**. Unless stated otherwise, all reagents were purchased from Sigma–Aldrich. The std20S (BML-PW8720–0050), i20S (BML-PW9645-0050), PA28γ (BML-PW9875-0100), and PA28αβ (BML-PW9420-0025) were purchased from Enzo Life Science. Deuterium oxide (D215H) and sodium deuteroxide (D076Y) solutions were from Euriso-top and deuterium chloride (543047-50 G) was from Sigma–Aldrich. The fluorogenic peptide substrate (4011369.0025) for proteasome activity measurement was from Bachem.

**Proteasome activity test**. The assay was performed in black 96-well plates (Fluotrac 200—GREINER BIO-ONE). 5 μL of each sample (corresponding to 50 ng of 20S and/or 28 ng of PA28) were added in three independent experiments to 45 μL of Tris-HCl 100 mM pH 8 and 50 μL of Suc-LLVY-AMC (for chymotrypsin-like activity) substrate in 200 mM Tris-HCl, pH 8 at a final concentration of 400 μM/well. The kinetic assays were performed at 37 °C in a FLX-800 spectrofluorimeter (BIOTEK, Winooski, VT, U.S.A.) over 90 min with one reading every 5 min, at 360 nm for excitation and 460 nm for emission. Each proteasome specific activity was

obtained by dividing the activity by the proteasome quantity (50 ng). A two-sided $t$-test was used between the std20S alone and in presence of PA28.

**Development of a HDX-MS pipeline dedicated to the comparative analysis of 20S proteasome/regulator complexes**. In a classical HDX-MS workflow, the proteins of interest are digested online and the generated peptides are separated on a reverse phase column before MS analysis to monitor deuterium incorporation rates. The main challenges encountered when analysing heterogeneous complexes such as the proteasome by HDX-MS are threefold. First, the high number of peptides generated after digestion entailed specific optimization of their chromatographic separation. Second, we optimized the quenching step and injection parameters to reduce dead volumes, as well as the acquisition method, in order to handle samples at low concentration (<1 μM). Finally, we developed a computational pipeline dedicated to the multidimensional analysis of HDX-MS analysis of large complexes. The resulting data were mapped to available 3D structures using our recently developed open-source web application HDX-Viewer v.1.2[67].

**Automated Hydrogen-Deuterium eXchange coupled to Mass Spectrometry (HDX-MS)**. HDX-MS experiments were performed on a Synapt-G2Si (Waters Scientific, Manchester, UK) coupled to a Twin HTS PAL dispensing and labelling robot (LEAP Technologies, Carborro, NC, USA) via a NanoAcquity system with HDX technology (Waters, Manchester, UK). Data were collected with MassLynx v.4.1 from Waters Scientific (Manchester, UK) and the robot was controlled via HDx Director v.1.0.3.9 (LEAP Technologies, Carborro, NC, USA). Each step was optimized to minimize sample loss and work with such a heterogeneous and diluted sample:

*Method in HDxDirector*: The method recommended by Waters (HDX System Suitability Test) injects only 25% of the sample that is aspirated from the protein vial. In order to reduce sample loss, we (1) used a sample loop of 100 μl instead of 50 μl, (2) carefully reduced all the dead volumes, and (3) used 500 mM glycine pH 2.3 instead of 50 mM $K_2HPO_4$, 50 mM $KH_2PO_4$, pH 2.3 in order to optimize the ratio of quenching volume (10% instead of 50%). With this workflow, we increased by more than threefold the amount of starting material injected (79%). 20S proteasomes were incubated alone or with a 2-fold molar excess of PA28, with final concentrations of 0.4 and 0.8 μM, respectively. 5.7 μL of protein were aspirated and 5.2 μL were diluted in 98.8 μL of protonated (peptide mapping) or deuterated buffer (20 mM Tris pH/pD 7.4, 1 mM EDTA, 1 mM DTT) and incubated at 20 °C for 0, 0.5, 1, 5, 10, and 30 min. The final $D_2O$ percentage was 95% in the labelled samples. 99 μL were then transferred to vials containing 11 μL of precooled quenching solution (500 mM glycine at pH 2.3). For experiments involving PA28αβ, the quenching buffer was supplemented with 250 mM tris-(2-carboxyethyl) phosphine (TCEP) in order to reduce the disulphide bridge between Cys21 of chain α and Cys3 of chain β. After 30 s. of quenching, 105 μL were injected into a 100 μL loop. Proteins were digested online with a 2.1 × 30 mm Poros Immobilized Pepsin column (Life Technologies/Applied Biosystems, Carlsbad, CA, USA). The temperature of the digestion room was set at 15 °C.

*Chromatographic run*: In order to cope with the unusual sample heterogeneity, the runtime of the chromatographic separation (12 min) was doubled compared to the one used for smaller protein complexes (6 min). Peptides were desalted for 3 min on a C18 pre-column (Acquity UPLC BEH 1.7 μm, VANGUARD) and separated on a C18 column (Acquity UPLC BEH 1.7 μm, 1.0 × 100 mm) by the following gradient: 5–35% buffer B (100% acetonitrile, 0.2% formic acid) for 12 min, 35–40% for 1 min, 40–95% for 1 min, 2 min at 95% followed by 2 cycles of 5–95% for 2 min and a final equilibration at 5% buffer A (5% acetonitrile, 0.2% formic acid) for 2 min. The total runtime was 25 min. The temperature of the chromatographic module was set at 4 °C. Experiments were run in triplicates and the protonated buffer was injected between each triplicate to wash the column and avoid cross-over contamination.

*MS acquisition*: The acquisitions were performed in positive and resolution mode in the m/z range 50–2000 Th. The sample cone and capillary voltages were set at 30 and 3 kV, respectively. The analysis cycles for nondeuterated samples alternated between a 0.3 s low energy scan (Trap and Transfer collision energies set to 4 V and 2 V, respectively), a 0.3 sec high energy scan (Ramp Trap and Transfer collision energies set to 18 to 40 V and 2 to 2 V, respectively) and a 0.3 sec lockspray scan (0.1 μM [Glu1]-Fibrinopeptide in 50% acetonitrile, 50% water and 0.2% formic acid infused at 10 μL/min). The lockspray trap collision energy was set at 32 V and a GFP scan of 0.3 sec is acquired every min. In order to double the signal intensity of deuterated peptides, deuterated samples were acquired only with the low energy and lockspray functions.

*Data analysis*: Peptide identification was performed with ProteinLynx Global SERVER v.3.0.2 (PLGS, Waters, Manchester, UK) based on the MS$^E$ data acquired on the nondeuterated samples. The MSMS spectra were searched against a home-made database containing sequences from the 17 std20S and i20S subunits, PA28α, PA28β, PA28γ, and pepsin from *Sus scrofa*. Peptides were filtered in DynamX v.3.0 from Waters Scientific (Manchester, UK) with the following parameters: peptides identified in at least two replicates, 0.2 fragments per amino-acid, intensity threshold 1000. The quantitative analysis of deuteration kinetics was performed using the statistical package R (R Development Core Team, 2012; http://www.R-project.org/) on the corresponding MS intensities. The deuterium uptakes of each

ion for each timepoint were calculated based on the theoretical maximum, considering that all amino-acids (except proline residues and the first amino-acid or each peptide) were deuterated, and then averaged (weight = intensity) to get a value of relative deuterium uptake (RDU) per peptide sequence/condition/timepoint (see Supplementary Data 4, 5, 12, 14, 16). The RDUs were not corrected for back exchange. To identify the protein regions that presented conformational changes in complex vs. alone, we performed an ANOVA (Anova(), type = III, singular.ok = T) followed by Benjamini–Hochberg correction of the $P$ value on all quantified peptides. For each comparison, we considered significantly regulated the peptides with a corrected $P$ value $\leq 0.01$ and an absolute difference of RDU above 1.57% (four times the mean absolute difference of RDU in the entire dataset) for three successive time points (Supplementary Data 9–10). The corresponding volcano plots are presented in Supplementary Data 3, 7, 13 and all the regions statistically regulated in all the differential HDX-MS analysis are listed in Supplementary Data 8. The RDU and differences of RDU (for protein alones or comparison between conditions, respectively) were consolidated (RDU mapping from peptide-level to protein sequence) using the values of the smallest peptide to increase the spatial resolution (see consolidation heatmaps in Supplementary Data 1–3, 6, 11). These data (per timepoint or sum of all time points) were then used for structural data mining with the recently developed open-source web application called HDX-Viewer [https://masstools.ipbs.fr/hdx-viewer][67] that allows to directly plot and visualize the whole dataset (14 different subunits per proteasome core particle) on the proteasome 3D structure. The scripts corresponding to the quality control and statistical analysis of this dataset are available at zenodo.org [https://zenodo.org] with the https://doi.org/10.5281/zenodo.3769174 under the Creative Commons Attribution 4.0 International licence. All molecular representations were generated in UCSF ChimeraX v.0.9 (2019–06–06)[68].

*Modeling*: The PA28γ model was generated using the structure of human PA28α (PDB:1AV0) as a template with the SWISS-MODEL server[52]. This model misses the first 3 and the last 7 amino-acids, as well as the 64–106 stretch.

**Reporting summary**. Further information on research design is available in the Nature Research Reporting Summary linked to this article.

## Data availability

The mass spectrometry proteomics data have been deposited to the ProteomeXchange Consortium via the PRIDE[69] partner repository with the dataset identifier PXD018921. All other data are available with this paper online.

## Code availability

The R scripts of the analysis, with the fasta files and output tables and cxc files are freely available at zenodo.org [https://zenodo.org] with the https://doi.org/10.5281/zenodo.3769174 under the Creative Commons Attribution 4.0 International licence.

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

## Acknowledgements

This work was supported by the French Ministry of Research (ANR-ProteasoRegMS to J. M. and Investissements d'Avenir Program, Proteomics French Infrastructure, ANR-10-INBS-08 to O.B.-S.), the 631 Fonds Européens de Développement Régional Toulouse Métropole and the Région Midi-Pyrénées (O.B.-S.), and the Novo Nordisk Foundation (NNF14CC0001).

## Author contributions

All authors contributed to editing the manuscript and figures. J.L. performed the HDX-MS optimization and data acquisition, model building and data analysis. J.P. performed the HDX-MS optimization, data acquisition, and data analysis. D.Z. and T.M. performed experiments. M.L.P. analyzed the data and wrote the paper. O.B.-S. provided critical input. M.P.B. provided critical expertise on the proteasome. J.M. designed, performed and supervised the experiments, data and structural analysis, and wrote the paper.

## Competing interests

The authors declare no competing interests.
