## [Peer Review File · Nature Communications]

REVIEWER COMMENTS

Reviewer #1 (Remarks to the Author):

The 20S proteasome is a multi-catalytic large protease complex made up of four axially stacked heteroheptameric rings. In eukaryotes, two outer α -rings and two inner β -rings are formed by seven homologous but distinct $\alpha 1$ – $\alpha 7$ and $\beta 1$ – $\beta 7$ subunits, respectively. Of the seven β -subunits, $\beta 1$, $\beta 2$, and $\beta 5$ have proteolytic activities. Along with the standard 20S proteasome (std20S), it has been known that the immune 20S proteasome (i20S) exists in mammalian cells, in which the catalytic subunits are replaced with the specific subunits, $\beta 1i$, $\beta 2i$, and $\beta 5i$. These 20S proteasomes themselves are basically latent because the N-termini of α subunits occupies the centers of the α -rings. For opening the closed gates of the α -rings, there are various activators, such as the 19S particle, PA28 $\alpha\beta$, PA28 γ , and PA200. Binding of these activators to the CP changes the status of the N-termini of α subunits, although the mechanism is different from each other. Also, it is well known that the binding of the activators modulates the 20S peptidase activities. This implies that binding of the activators to the α -rings leads to the conformational changes of not only the α subunits but also the β subunits. However, the precise structural evidence has not reported.

In this report, Hydrogen-Deuterium eXchange coupled to Mass Spectrometry (HDX-MS) was used for the first time to investigate conformational differences between the human std20S and i20S alone and in complex with PA28 $\alpha\beta$ and PA28 γ . Although we cannot obtain detailed structural information by HDX-MS, like NMR, x-ray, and cryo-EM analyses, this new approach is interesting and informative to know the dynamic changes of the 20S by the incorporation of the immune-subunits and the binding of PA28. This reviewer has just one major concern (point #1) and some minor concerns listed below.

1. It is surprising for this reviewer to know that binding of PA28 γ did not affect any of the std20S α subunit N-termini (Fig. 7F). The authors should confirm whether their std20S-PA28 γ complex has higher peptidase activities than their std20S alone by biochemical experiments. Is it possible that the binding of PA28 γ opens the gate without affecting any of the α subunit N-termini? Also, this reviewer thinks that "less PA28 γ -driven opening of the std20S than the i20S pore" is correct in line 267, judging from Figs 7F and 7P, although this conclusion seems contradictory to the previous findings described in line 286.
2. It would be helpful to show where the dynamic patches and the 1-2, 3-4, and 5-6 α -pockets are located in Fig. 2C, described in lines 127-130.
3. This reviewer does not understand the reason why the sequence coverage of the immune-specific subunits was poorer than the standard subunits. Could the authors provide a possible reason for this result?
4. What are the numbers of 57-98, 70-98, and 64-106 in Fig. 5B? Do these numbers represent the regions of the apical loops in each PA28?
5. Please define the red circles in Figs. 7N and 7S.
6. This reviewer cannot appreciate the differences between the solvent accessibilities of $\beta 1i$ active sites circled in orange in Figs. 7N and 7S, although the authors claim that the solvent accessibility of $\beta 1i$ active site is increased by binding of PA28 $\alpha\beta$ in line 284.
7. Fig. 5D should be Fig. 4D in line 327, and Fig. 5C should be Fig. 4C in line 329.

Reviewer #2 (Remarks to the Author):

The present study on proteasome structural dynamics using HDX-MS is very impressive and must have been a Herculean effort to carry out, given the size and complexity of this important protein assembly. I commend the authors for this excellent work and expect the data to form a valuable basis for future work also by the community. A similar approach will undoubtedly also be informative when applied to conformational studies of other large and dynamic macromolecular complexes in the cell. I recommend this manuscript for publication and have only minor comments which I would like the

authors to address.

Probably the most important question I would like the authors to comment on in more detail is about the likely conformational heterogeneity - are co-existing conformational poses resolved, and are "minority" states able to be detected in these data?

Some of the uptake curves show interesting (albeit not entirely uncommon) behaviour, such as maxima/minima, or e.g. crossover points. To what extent do these features represent actual conformational behaviour vs. being artefactual (within error) or highlighting protein stability issues?

page 5, line 123: How far is an assumption made that the most flexible sequences are also the most accessible? HDX ultimately samples degree of H bonding so the two are not necessarily the same.

page 17, line 420ff: Why is glycine considered a stronger acid? What final percentage D2O were the experiments carried out in? Was any back-exchange correction applied?

figure 1: please explain the comment "not reported in the heat map also covered by smaller peptide" as not entirely clear what it means - was peptide (sequence) overlap exploited here?

figure 4 legend line 691: best to emphasize here again which way around the Delta is meant between the two states

Frank Sobott

Reviewer #3 (Remarks to the Author):

Review of "Conformational maps of human 20S proteasomes reveal PA28- and immuno-dependent inter-ring crosstalks", Lesne et al 2020

Overall comments

In this paper, the authors have performed HDX-MS experiments to identify the differences between the standard and immuno-20S proteasomes, with and without the proteasome regulator complexes PA28a/b and PA28g. By analyzing the changes in solvent accessibility of the various complexes, the authors determined that the 20S proteasome catalytic b-ring, allosterically induces changes in the a-ring, and that this varies between the standard and immune proteasomes. Upon binding of the PA28 regulators to the a-rings, changes were observed in the b-rings, an effect that varied depending on the regulator-proteasome pairing.

The data presented is interesting and important for our understanding of the dynamism of the 20S proteasome and its regulation. However, explanations of the data are either lacking in detail or difficult to follow, and the discussion of the data in the context of our current understanding of proteasome structural dynamics is insufficient and full with speculations.

A considerable revision of the manuscript is warranted before acceptance for publication.

Major comments

1. HDX reveals the extent of solvent exposure not necessarily stability, flexibility or instability. The direct link in the manuscript between RDU and structural features is not clear.
2. All the data are collected along a time series, but there is no analysis of the rate? Relevance? How is it used to provide information of the RDU? Is there global change? Does the time domain say anything about the dynamics?
3. It is not clear how the 20S proteasome becomes more solvent accessible upon PA28 interaction, while the PA28 complexes becomes less. This interpolation needs to be backed up by additional experimental methods.
4. In the Discussion (page 14-15) it is stated that protection cannot be correlated to affinity. However, difference in affinity are discussed all along the Result section.

5. Page 4, line 108: Explain the workflow in more detail, refer to specifics outlined in Figure 1A.
6. Page 4, line 109-112: Explain the consolidation and differential analyses in more detail, refer to specifics outlined in Figure 1B-E.
7. Along the manuscript changes in RDU are considered to be significant, however numbers are not provided to indicate significance. A table showing significant changes should be added where relevant.
8. No information is provided about the two antechambers. Is there more solvent exposure when the pore is open? Please discuss.
9. The binding of 20S and 20Si to the PA28 regulators is very confusing as there is no obvious pattern. Therefore, supporting experiments are required to rule out technical issues.
10. The suggested higher affinity of 20S to PA28ab than to PA28g should be validated by an additional method.
11. Prove activity and purity of the purchased complexes.
12. Page 6, lines 153-155: How much more deuterated is T20 and T22 of B1/B1i? State the numbers as by eye in the figure it doesn't look very different. Is it a significant difference? Same goes for the S1 pocket of B5/B5i, list the relevant amino acids (A20/V31?) and their RDU in the text. Can the conclusion for more flexibility in the active sites/S pockets really be drawn for a small difference in RDU for one or two residues? If yes then it needs to be clearly justified.
The section title is misleading, as stated in the text the direct comparison between the STD and i20S catalytic subunits was not possible due to poor sequence coverage, therefore no statement can be made on any conformational difference between the two proteasomes. Thus, the concluding sentence is also misleading, as there is not enough of a difference in their flexibility. If this is not the case then explain in more detail.
13. Figure 1: In general the figure is very busy, can it be simplified?
A: Describe the workflow in more detail in the legend.
B: "Protein sequence" should refer to the amino acid letter sequence that is directly above the heatmap, "residue number" should be labelled above the arrow
Describe in the legend what the heatmap colours mean i.e. light blue is less RDU, dark blue is more RDU? In general across all the figures RDU changes from a percentage to a decimal number.
D: Describe in the legend the statistical analysis outlined
Is there correlation between the long and small peptides? Explain the rationale for choosing only the smaller peptides.
14. Figure 2: The regions highlighted with red arrows in C-F are not specifically mentioned in the text as stated in the legend. Describe their significance in the text or in the legend.
B: What is the red circle highlighting? Not described in the legend
15. Page 5, lines 128-133: The patches on the interface that correspond to the pockets for the RPT tails should be highlighted in Figure 2 for easy identification. Also recommended to elaborate on the regions highlighted in with red arrows in the figure, what is their relevance?
16. Figure 3: Why not include the differential analysis of the catalytic subunits here to make the differences between the STD and i20S more obvious? Or expand Figure 4 to highlight the differences.
17. Figure S2, S6 and S10: Provide the same y axis for all graphs to allow a quicker comparison between the RDUs of each peptide possible.
18. Page 7, line 168: As mentioned in the previous section, the differences in the catalytic and substrate pockets between the STD and i20S were not significant, so how can this translate to significant conformational changes in the non-catalytic subunits?
19. Figure 5: A: Point out the missing residues 15-31 in the structure in 5A, is this the dashed line? Include a cartoon representation of the entire heteroheptamer, showing the position of the apical loops
B: The RDU scale is 0-250%, but in Fig S7, from which these structures are based, the RDU goes from 0-100% (0.0-1.0)? By eye the blue color of the RDUs don't match up to what is represented in the structures, in B-D.
C-D: Include arrows showing direction of rotation between the structures
D: State in legend that this is a homology model.
20. Page 8, line 211-212: Elaborate on the significance of the activation loop of PA28B being less deuterated, as this is the title of the section it should be explained in more detail. Why is this

important?

21. Figure 6: State in the legend that the color coding indicates that blue is less accessible in the complex and red is more accessible in the complex. Include arrows to show direction of rotation between the images. Change the colors of the black arrows pointing at the activation loops, they are hard to see. Include the differential analysis images of the PA28 with the i20S, in Fig S8 it is clear that the effect is the opposite, i.e. the PA28 subunits become more accessible upon i20S binding. This is briefly mentioned on page 9, lines 235-238, elaborate on this, it is interesting.
22. Page 9, line 220: These listed residue regions are not highlighted in the figure, and in the figure there are residue regions highlighted in red that are not mentioned in the text?
23. Page 9, line 225: State the residue numbering of the kinks. Why are they indicated?
24. Figure 7: This is one of the most important results but it not clear at all. In general, the description in the text of the results in Figure 7 is extremely difficult to follow and is confusing. Perhaps representing the data in a different more manageable way is warranted. For example, include larger labels for the columns of structures at the top of the figure, indicating which columns correspond to STD and i20S, combined with which PA28 regulators. Explain in the legend what the red circles are referring to.
25. Page 10, line 247-249: State the differences in deltaRDU calculated for the C-ter of a3 depicted in Fig7A and Fig 4D, by eye the delta RDU looks like it indicates a strongly protected region in both analyses.
26. Page 10, line 251: the regions in 7A that are being referred to are blue from the analysis but circled in red? If there are circles in the figure refer to them, or leave them out.
27. Discussion: In general, the discussion reads more like a summary of the results section, and not a discussion of the results in the context of the literature. A summary of the results is definitely warranted, although perhaps this can be done briefly at the beginning of the discussion section, and not take up the majority of it.
28. Figure 8: The legend needs to be much more descriptive. This is the model of your major findings, explain in much greater detail.
A: Colors in the text are referred to as orange and purple?
B: The text describes this as showing stabilization of the loops, this isn't clear in the figure. Also, what are the + and - symbols representing?
C: What are the - symbols representing?
29. Methods: Page 16, lines 396-398: Include catalog numbers for the listed reagents.
30. Page 10, line 259: Which red circles in 7C are being referred to? There are 3...
31. Page 10, line 262: There are no blue circles in 7C, only red circles. Clarify which region is being referred to.
32. Page 11, line 266-267: Stated that the binding of PA28g only affected a2 and a3 Nterminals of the i20S, and yet in 7P it clearly looks like the a1 Nter is also destabilized, even more than the a2?
33. Page 11, line 273: Where in 7F is the a2 destabilization? Why is it not circled like the other regions being discussed in 7H?
34. Page 11, line 282-283: Include the RDU numbers to demonstrate the significant changes upon PA28ab binding as by eye it doesn't look convincing. Perhaps a small table with the relevant numbers in the figures throughout the manuscript that are significant would help.
35. Page 11, line 284: The B1i active site in 7N does not look significantly affected, in fact most of the active site circled in orange is gray.
36. Page 12, line 289-290: What are the numbers to say there is a significant decrease? Is the B2i N-terminal not visible in Fig 7S? State this if this is the case.
37. Page 12, line 312: First mention of residues 238-261, where in the results section is this described?
38. Page 13, line 313: There is no purple coloring in Figure 8.
39. Page 13, line 323-326: Elaborate on the comparison with Pf20S, why is this species being mentioned? Is this one of the few structures of 20S bound to PA28?
40. Page 13, line 328: There is no orange coloring in Figure 8.
41. Page 13, line 331: There are no orange arrows in Figure 8.

42. Page 14, line 359: Figure 8B doesn't seem to be showing stabilization of the loops?

Additional comments

43. Page 2, lines 42-48: Include references for the UPS and its functions.

44. Page 2, lines 51-55: Include more detail about the catalytic subunits i.e. what are their different enzymatic activities

45. Page 3, line 72: Other citations needed for additional PIPs beyond Ecm29

46. Page 4, line 90: Correct P28 to PA28

47. Page 4, line 90-92: Elaborate on the HDX-MS method, a more detailed explanation is required. Also include references for the method.

48. Page 6, lines 159-161: 'mice' should be 'mouse'. This sentence is confusing. Reword to make it clearer.

49. Page 6, lines 147: 'remaining' should be 'remainder'

50. Page 8, line 190: "An activation loop is less dynamic in PA28B"

51. Page 8, line 200: Define Pf, and explain why this species sequence is being used for comparison

52. Page 8, line 202: Include a more recent reference for the SWISS-MODEL server.

53. Page 10, line 252: 'remaining' should be 'remainder'

54. Page 10, line 255: State that all the regions being discussed in 7A are circled in red.

55. Page 10, line 258: "...and is in close vicinity to the B5 interface, which is also more dynamic"

56. Page 13, line 335: PGPH activity should be referred as caspase-like activity. Also "than the one of" should read "than that of"

Please find below the point-by-point answer to all the reviewers' comments.

REVIEWER COMMENTS

Reviewer #1 (Remarks to the Author):

The 20S proteasome is a multi-catalytic large protease complex made up of four axially stacked heteroheptameric rings. In eukaryotes, two outer α -rings and two inner β -rings are formed by seven homologous but distinct $\alpha 1$ – $\alpha 7$ and $\beta 1$ – $\beta 7$ subunits, respectively. Of the seven β -subunits, $\beta 1$, $\beta 2$, and $\beta 5$ have proteolytic activities. Along with the standard 20S proteasome (std20S), it has been known that the immune 20S proteasome (i20S) exists in mammalian cells, in which the catalytic subunits are replaced with the specific subunits, $\beta 1i$, $\beta 2i$, and $\beta 5i$. These 20S proteasomes themselves are basically latent because the N-termini of α subunits occupies the centers of the α -rings. For opening the closed gates of the α -rings, there are various activators, such as the 19S particle, PA28 $\alpha\beta$, PA28 γ , and PA200. Binding of these activators to the CP changes the status of the N-termini of α subunits, although the mechanism is different from each other. Also, it is well known that the binding of the activators modulates the 20S peptidase activities. This implies that binding of the activators to the α -rings leads to the conformational changes of not only the α subunits but also the β subunits. However, the precise structural evidence has not reported.

In this report, Hydrogen-Deuterium eXchange coupled to Mass Spectrometry (HDX-MS) was used for the first time to investigate conformational differences between the human std20S and i20S alone and in complex with PA28 $\alpha\beta$ and PA28 γ . Although we cannot obtain detailed structural information by HDX-MS, like NMR, x-ray, and cryo-EM analyses, this new approach is interesting and informative to know the dynamic changes of the 20S by the incorporation of the immune-subunits and the binding of PA28. This reviewer has just one major concern (point #1) and some minor concerns listed below.

1. It is surprising for this reviewer to know that binding of PA28 γ did not affect any of the std20S α subunit N-termini (Fig. 7F). The authors should confirm whether their std20S-PA28 γ complex has higher peptidase activities than their std20S alone by biochemical experiments. Is it possible that the binding of PA28 γ opens the gate without affecting any of the α subunit N-termini?

We thank Reviewer #1 for this comment and we are sorry that this was not detailed further in the first version of the manuscript.

To validate the complex formation, we checked that both PA28 regulators utilized in this study were able to activate the std20S proteasome using the same ratio as in the HDX-MS experiment (2-fold molar excess of PA28 regulator), as shown in the Figure below. Thus, the absence of significant protection of the std20S α subunit N-termini cannot be explained by a lack of PA28 activity/complex formation.

Supplementary Fig. S8: Proteolytic activity test (chymotrypsin-like) of the std20S alone and in presence of PA28 γ or PA28 $\alpha\beta$. Mean of protease activity relative to control (std20S) in 3 independent experiments. The error bars correspond to the standard error, the *P*-values of a paired two-sided t-test against the std20S alone are presented above the corresponding bars.

We are now discussing this point on line 372 of the manuscript:

“The very limited decrease of RDU on the α -ring interface interacting with PA28 regulators can be explained by the resulting gate opening (RDU increase) known to induce its increased proteolytic activity. Gate opening and binding interface protection are expected to have opposite impacts on RDU, so their combination in the same protein region would result in no significant changes of deuteration rates.”

The results of the PA28-driven activation of the std20S proteasome are now commented on line 214: *“Before comparing the PA28 regulators alone and in complex with the std20S, we confirmed their ability to increase the proteasome activity in vitro. In our hands, a 2-fold molar excess of any of the PA28 regulators doubled the std20S chymotrypsin-like activity (P-value of a paired two-sided t-test: 0.0007 and 0.0013 for PA28 γ and PA28 $\alpha\beta$, respectively; Fig. S8 and Table S6)”* and included as **Supplementary Fig. S8** and **Table S6**. The numerotation of the supplementary figures and tables was modified accordingly and the details of the proteasome activity test are now described in the Methods section.

Also, this reviewer thinks that "less PA28 γ -driven opening of the std20S than the i20S pore" is correct in line 267, judging from Figs 7F and 7P, although this conclusion seems contradictory to the previous findings described in line 286.

We thank Reviewer #1 for spotting this mistake. The sentence “less PA28 γ -driven opening of the i20S than the std20S pore” should on the contrary read “**less PA28 γ -driven opening of the std20S than the i20S pore**”, since two N-termini are destabilized by PA28 γ on the i20S and none on the std20S. We apologize for this mistake and have corrected the text accordingly.

2. It would be helpful to show where the dynamic patches and the 1-2, 3-4, and 5-6 α -pockets are located in Fig. 2C, described in lines 127-130.

We thank the reviewer for this suggestion. We modified the text in lines 127-130 to include the sequences of the corresponding peptides. These were already labelled in red in Fig. 2C. As also requested by Reviewer #3 (comment #15), we circled in white the 7 α -pockets, as stated in the legend: “**The α -pockets (docking sites for the C-termini of PA28 regulators) are circled in white in (C).**”

3. This reviewer does not understand the reason why the sequence coverage of the immune-specific subunits was poorer than the standard subunits. Could the authors provide a possible reason for this result?

This is a very good question. It is expected to have variations of sequence coverage due to sequence differences, but not as much as the ones we observe between the standard and immuno-catalytic subunits. We think that this results from a slight contamination of the commercial i20S from Enzo that also contains residual std20S, as we recently reported using Top-Down MS⁶¹. Other commercial i20S from a different vendor (Boston Biochem) also revealed the presence of traces of std20S (Gersch et al, 2015). As a result, our i20S samples contain relatively less i20S than there is std20S in the std20S samples, which could explain why we detect less i20S-specific peptides. We now describe further the sample we worked with in the Results:

“For this, we purchased commercial i20S purified from human spleen previously used for in vitro proteolytic assays⁴²⁻⁴⁴.”

And in the Discussion:

“As commented before, the differential analysis of the catalytic subunits is impossible due to their sequence differences (the peptides resulting from the pepsin digestion are different). The comparison of their deuteration profiles was further hindered by the limited sequence coverage of the i20S. This was partly due to the presence of residual traces of std20S present in this sample⁶¹.”

4. What are the numbers of 57-98, 70-98, and 64-106 in Fig. 5B? Do these numbers represent the regions of the apical loops in each PA28?

They indeed represent the numbers of the residues from the apical loops that are missing in the crystal structures/models. We modified the figure: the apical loops are now also represented as dashed lines in Fig. 5B and we included a cartoon representation of the heptameric complex, as suggested by Reviewer #3 (comment #19). This is now described

in the figure legend: *“The residues corresponding to the missing apical loops (dashed lines) are indicated in (B). (C) Structure of the heptameric complex PA28 α 4 β 3 (PDB: 5MX5). Only 4 apical loops out of 7 are represented, for the sake of clarity”.*

5. Please define the red circles in Figs. 7N and 7S.

We added to the Fig.7 legend: *“Regions discussed in the text are circled in red.”*
We also modified the text in line 278-279: *“[The complex formation]...resulted in remodeling of the β -ring/ β -ring interface eventually affecting β 1i/2i/7 or β 1i/2i/5i/6/7 with PA28 $\alpha\beta$ (Fig. 7N, red circles on β 1i/2i/7) and PA28 γ (Fig. 7S, red circles on β 1i/2i/5i/6/7), respectively.”*

This was also requested by Reviewer #3 for Fig.7A (comment #54)

6. This reviewer cannot appreciate the differences between the solvent accessibilities of β 1i active sites circled in orange in Figs. 7N and 7S, although the authors claim that the solvent accessibility of β 1i active site is increased by binding of PA28 $\alpha\beta$ in line 284.

In line 284, we refer to the peptide 10-17 of β 1i, encompassing the catalytic residue D17, that is significantly more deuterated in the presence of PA28 $\alpha\beta$ (corrected P-value = $3.89 \cdot 10^{-3}$, sum of difference = 14%). The reviewer is right to say that this peptide is not visible on Fig. 7N, because it is masked by peptides 18-24 and 162-175. We thus modified the text to cite Fig. S11-S12 instead and added the statistics obtained for peptide 10-17 in the main text.

This question was also raised by Reviewer #3 (comment #35).

7. Fig. 5D should be Fig. 4D in line 327, and Fig. 5C should be Fig. 4C in line 329.

The text has been modified accordingly.

Reviewer #2 (Remarks to the Author):

The present study on proteasome structural dynamics using HDX-MS is very impressive and must have been a Herculean effort to carry out, given the size and complexity of this important protein assembly. I commend the authors for this excellent work and expect the data to form a valuable basis for future work also by the community. A similar approach will undoubtedly also be informative when applied to conformational studies of other large and dynamic macromolecular complexes in the cell. I recommend this manuscript for publication and have only minor comments which I would like the authors to address.

1. Probably the most important question I would like the authors to comment on in more detail is about the likely conformational heterogeneity - are co-existing conformational poses resolved, and are "minority" states able to be detected in these data?

This is an excellent question. Unlike cryo-EM-based classification that can literally freeze and resolve different coexisting conformations, HDX-MS is a solution-based method and the resolution of these states depends on the relative rates of exchange and rates of motion of the exchangeable protons. This is very well described in the literature (Hvidt and Nielsen, 1966; Kaltashov and Eyles, 2002). In certain cases the exchange is faster than the motion (EX1 kinetics) and this can be observed as two different isotopic motifs in HDX-MS. This is for example the case for certain SH3 domains (Wales and Engen, 2006) and ABC transporters (Li et al, 2018).

However, in most cases, the exchange is slower than the motion (EX2 dynamics) and the HDX-MS analysis of a mix of two conformational states *in solution* results in the averaged deuteration rates of both conformations. In the context of this study –comparison of two different samples (*i.e.* proteins alone vs. with its binding partner)—, it is possible to identify and localize relevant changes in solvent accessibility even if we work with a mixture of different conformations. However, these are averaged changes, and represent the majority of conformational changes occurring in the samples. For example, if incubation of proteins A and B lead to a homogeneous sample of 100% of heterodimer AB, the binding interface would be significantly less deuterated in A+B than in A and B analysed alone. If the incubation of A and B lead to 50% dimer only, the differences of average deuteration rates between the two conditions would drop. We would probably still observe the same interface, albeit attenuated by the presence of 50% of protein alone. So Reviewer #2 is right, the deuteration rates measured by HDX-MS are strongly impacted by the presence of co-existing conformational states in the sample. The quantity of complex formed in our samples (with a 2-fold molar ratio of PA28) was sufficient to activate the proteasome and detect subtle changes in solvent accessibility upon complex formation (**Fig. S8** added upon Reviewer #1's request), but "minority" states cannot be detected here as we did not identify any EX1 kinetics.

This is now discussed in the discussion (l.308): ***"It is important to note that, except for a few specific cases^{58,59}, HDX-MS does not inform on the presence of multiple conformational states in solution since it is based on the measured average of their***

RDU. Other approaches such as cryo-EM-based classification would be needed to better characterise intermediate or minor states. Nevertheless, comparing the same sample (like std20S) in presence or absence of regulator and performing thorough statistical analysis identified...

2. Some of the uptake curves show interesting (albeit not entirely uncommon) behaviour, such as maxima/minima, or e.g. crossover points. To what extent do these features represent actual conformational behaviour vs. being artefactual (within error) or highlighting protein stability issues?

As Reviewer #2 mentioned, some of the mean deuteration kinetics present unexpected shapes. Indeed, hydrogen to deuterium exchange should not result in crossover point or decreasing deuteration uptakes. That being said, it is classically observed in such study and results from experimental variance (sample preparation, MS acquisition and signal integration).

We differentiate two types of unusual behaviour:

- Crossover points between RDUs in two conditions:

The direct comparison of std20S vs i20S shows 2 curves that are not perfectly flat and present a crossover point, even if their isotopic motifs are really nice. The differences observed between the two curves are in the range of experimental variance (see overlapping error bars). These can be considered as technical noise and are hardly visible in the new **Fig. S2** requested by Reviewer #3 (with scaled y-axis). In our work, such peptides are ignored since there is no significant difference between the two conditions.

- Reproducible drop in RDU:

Raw MS signal for the peptide ANDGVLL of PSMA4:

In this second example (peptide 39-45 of PSMA4), there is what seems to be a reproducible decrease (red line: std20S), which can be attributed to MS signal integration. The isotopic envelopes in two different time points of the std20S are presented below:

Isotopic envelope for 5 min:

Isotopic envelope for 10 min (after dropping point):

The isotopes selected for quantification are presented in blue. There are 5 isotopes selected at 5 min, and only 3 at 10 min, which explains the drop in signal. In this example there is no grey line for the missing isotopes at 10 min, which means that the software was not able to recognise the shoulder at 704.395 as a peak (because less resolved etc...). So here, with the tools we had, we could not include these “missed” isotopes. Such error in signal integration is inherent to the technology (at least nowadays), and not systematically corrected by manual curation.

In this work, we performed a stringent statistical analysis at the peptide-level. Not only did we perform an ANOVA to keep only the peptides with a corrected P -value ≤ 0.01 (classical approach), we also filtered out the peptides that did not present an absolute difference of RDU strictly above 4 times the standard deviation in 3 consecutive time points. This removed from the final results the peptides with unexpected behaviors such as the ones described above. Indeed, these two examples were not considered statistically significant and they are not discussed in the manuscript.

We provide the RDU kinetics for all the peptides, as well as the numeric values for each condition and time points in the supplementary table. This way, the readers can assess the quality of the data if they are interested in a specific protein region. We first displayed the close-up versions of the uptake curves (*i.e* showing the min and max values for each peptide in Supplementary Fig. S2, S6 and S10). Following Reviewer #3’s suggestion, we now provide the same y -axis for all graphs in order to allow a quicker comparison between the RDUs of each peptide. In this new presentation (more usual in the field), the artefactual behaviors that are discussed in this response are not as visible as in the previous version.

3. REDACTED page 5, line 123: How far is an assumption made that the most flexible sequences are also the most accessible? HDX ultimately samples degree of H bonding so the two are not necessarily the same.

This comment can be linked to the first comment of Reviewer #3: “HDX reveals the extent of solvent exposure not necessarily stability, flexibility or instability. The direct link in the manuscript between RDU and structural features is not clear”.

Reviewer #2 is right to point out that the rates of H to D exchange depend not only on the accessibility of the protons but also on their engagement in secondary structures and more generally hydrogen networks (which can reflect a certain degree of flexibility).

We agree with both Reviewer #2 and #3 that this clarification was lacking and we now discuss it in more detail in the introduction: ***“The rate of exchange of each amide hydrogen depends on its solvent accessibility and involvement in the stabilization of secondary structures: regions that are relatively more solvent-accessible or flexible present a higher deuteration rate than the ones that are hidden in the core of the protein or rigid.”***

We were particularly cautious with our interpretation, which is why we tried to use as much as possible the terms “flexible/accessible” or “flexible/dynamic” throughout the text. However, in some cases, we indeed preferred to choose the term “accessible” (when talking about a solvent exposed surface), or “flexible/dynamic” (when talking about a loop

that seems unstructured from the X-ray structure). We believe this increases the manuscript readability, but we would be happy to reconsider upon request.

4. [redacted] page 17, line 420: Why is glycine considered a stronger acid?

We agree that the terminology used (stronger acid) is wrong, and we apologize for this mistake. The main point here is that the 500 mM glycine solution is able to reduce the pH of the labelling buffer with a reduced volume (quenching ratio) compared to the 50 mM phosphate buffer. We thus corrected ***“used a stronger acid (500 mM glycine pH 2.3 instead of 50 mM K₂HPO₄, 50 mM KH₂PO₄, pH 2.3)”*** with ***“used 500 mM glycine pH 2.3 instead of 50 mM K₂HPO₄, 50 mM KH₂PO₄, pH 2.3”***.

5. [redacted] What final percentage D₂O were the experiments carried out in?

As stated in the text, ***“5.2 μL [of protein] were diluted in 98.8 μL of...deuterated buffer”***, so the final D₂O percentage was 95%. This was added in line 426: ***“The final D₂O percentage was 95% in the labelled samples.”***

6. [redacted] Was any back-exchange correction applied?

No back-exchange correction was applied here, as commonly seen in recent high-impact publications, for eg. in (Faull et al, 2019): ***“The amount of relative deuterium uptake for each peptide was determined using DynamX (3.0) and are not corrected for back exchange”***. This is now clearly stated in line 467: ***“The RDUs were not corrected for back exchange”***.

7. [redacted] figure 1: please explain the comment "not reported in the heat map also covered by smaller peptide" as not entirely clear what it means - was peptide (sequence) overlap exploited here?

Other reviewers found that **Fig. 1** was not clear enough, and we apologize. Hence, we modified the figure and the legend to improve clarity. Furthermore, we believe we did not explain enough our strategy for consolidation (mapping of peptide RDUs to the protein sequences) in the main text. It is now described in more detail at the beginning of the Result session.

For this project, we decided not to exploit overlapping peptides and to report the RDU of the shortest peptide to the protein sequence because it is the value that best represents the accessibility of this protein region, and it was in our hands the most robust strategy. This is further detailed in the answer to Reviewer #3's comment #13.

8. figure 4 legend line 691: best to emphasize here again which way around the Delta is meant between the two states

We are sorry this was not clear in the previous version of the manuscript. The initial legend “*Sum of differential relative uptakes ($\Delta(RDU)$) between the *i20S* and the *std20S*” was modified to “*Sum of differential relative uptakes between the *i20S* and the *std20S* ($\Delta(RDU)=RDU_{i20S}-RDU_{std20S}$)””. Similarly, and for the sake of homogeneity, we added “($\Delta(RDU)=RDU_{PA28/20S}-RDU_{PA28}$)” and “($\Delta(RDU)=RDU_{20S/PA28}-RDU_{20S}$)” in **Fig. 6** and **Fig. 7**, respectively.**

Frank Sobott

Reviewer #3 (Remarks to the Author):

Review of “Conformational maps of human 20S proteasomes reveal PA28- and immuno-dependent inter-ring crosstalks”, Lesne et al, 2020

Overall comments

In this paper, the authors have performed HDX-MS experiments to identify the differences between the standard and immuno-20S proteasomes, with and without the proteasome regulator complexes PA28a/b and PA28g. By analyzing the changes in solvent accessibility of the various complexes, the authors determined that the 20S proteasome catalytic b-ring, allosterically induces changes in the a-ring, and that this varies between the standard and immune proteasomes. Upon binding of the PA28 regulators to the a-rings, changes were observed in the b-rings, an effect that varied depending on the regulator-proteasome pairing.

The data presented is interesting and important for our understanding of the dynamism of the 20S proteasome and its regulation. However, explanations of the data are either lacking in detail or difficult to follow, and the discussion of the data in the context of our current understanding of proteasome structural dynamics is insufficient and full with speculations.

A considerable revision of the manuscript is warranted before acceptance for publication.
Major comments

1. HDX reveals the extent of solvent exposure not necessarily stability, flexibility or instability. The direct link in the manuscript between RDU and structural features is not clear.

This comment can be linked to one of the comments of Reviewer #2: “How far is an assumption made that the most flexible sequences are also the most accessible? HDX ultimately samples degree of H bonding so the two are not necessarily the same”.

Deuteration rates do not only reflect solvent exposure, but they also depend on the implication of the amide protons in secondary structure stabilisation (hydrogen bonding), as explained in many HDX-MS reviews and papers (Deng et al, 2016; Goswami et al,

2015). This is for example very concisely explained by John Engen and coworkers (Fang et al, 2014): “The rate of HX is strongly dependent on hydrogen bonding and solvent accessibility and is therefore directly related to protein structure and dynamics”.

As such, an α -helix that is exposed to the solvent can exchange more slowly than an unstructured loop buried in the interior of the protein (Skinner et al, 2012).

We agree with both Reviewer #2 and #3 that this clarification was lacking and we now describe it better in the introduction: ***“The rate of exchange of each amide hydrogen depends on its solvent accessibility and involvement in the stabilization of secondary structures: regions that are relatively more solvent-accessible or flexible present a higher deuteration rate than the ones that are hidden in the core of the protein or rigid.”***

We were particularly cautious with our interpretation, which is why we used as much as possible the terms flexible/accessible or flexible/dynamic throughout the text. However, in some cases, we indeed preferred to choose the term accessible (when talking about a solvent exposed surface), or flexible/dynamic (when talking about a loop that seems unstructured from the X-ray structure).

2. All the data are collected along a time series, but there is no analysis of the rate? Relevance? How is it used to provide information of the RDU? Is there global change? Does the time domain say anything about the dynamics?

This is a good point. Although we performed time-resolved measurements we did not run any detailed analysis based on the kinetics/curve-fitting/rates analysis, etc... We are aware that this can be done –although it is far from being systematic– but chose not to do it, mainly because such analysis requires a longer deuteration time span with additional time-points including two or three points at equilibrium. This was hardly manageable due to the size of the dataset (number of conditions tested). However, the strong statistical analysis performed in this project allowed us to identify reproducible differences of deuteration rates between the conditions tested.

We are not sure what the question “is there global change” means. Maybe Reviewer #3 wonders if the peptides behave similarly across the time of the experiment? We did not observe a homogeneous response to deuteration, and we believe that this is visible in the supplementary figures and tables provided with the manuscript. We hope this answers Reviewer #3’s question.

3. It is not clear how the 20S proteasome 20S becomes more solvent accessible upon PA28 interaction, while the PA28 complexes becomes less. This interpolation needs to be backed up by additional experimental methods.

We understand that having protection on the binding interface of the regulators (PA28) but not on the opposite binding site (α -ring of the 20S) seems rather counter-intuitive.

This point was also raised by Reviewer #1 in his/her first comment, in which he/she asks for additional experiments to prove the activation of the 20S proteasome by the PA28 regulators.

To validate the complex formation, we checked that both PA28 regulators utilized in this study were able to activate the std20S proteasome using the same ratio as in the HDX-MS experiment (2-fold molar excess of PA28 regulator), as shown in our new Supplementary **Fig. S8**.

This is now included on line 220: ***“Before comparing the PA28 regulators alone and in complex with the std20S, we confirmed their ability to increase the proteasome activity in vitro. In our hands, a 2-fold molar excess of any of the PA28 regulators doubled the std20S chymotrypsin-like activity (P-value of a paired two-sided t-test: 0.0007 and 0.0013 for PA28 γ and PA28 $\alpha\beta$, respectively; Fig. S8 and Table S7)”***.

Thus, the absence of significant protection of the std20S α subunit N-termini cannot be explained by a lack of PA28 activity/complex formation.

We explain this point further on line 372 of the manuscript:

“The very limited decrease of RDU on the α -ring interface interacting with PA28 regulators can be explained by the resulting gate opening (RDU increase) known to induce its increased proteolytic activity. Gate opening and binding interface protection are expected to have opposite impacts on RDU, so their combination in the same protein region would result in no significant changes of deuteration rates.”

4. In the Discussion (page 14-15) it is stated that protection cannot be correlated to affinity. However, differences in affinity are discussed all along the Result section.

This statement was written in the context of the unexpected small impact of PA28 binding to the proteasome 20S solvent-facing α -ring (see previous comment). Although protein binding can induce protection to deuteration, the latter cannot systematically be correlated to affinity (as protein binding can sometimes induce a destabilization of the binding interface that can compensate for the steric hindrance of the protons). We believe this was unclear in the previous version of the manuscript and thanks to Reviewer #3's comment, we explained further this point line 372 of the new version:

“The very limited decrease of RDU on the α -ring interface interacting with PA28 regulators can be explained by the resulting gate opening (RDU increase) known to induce its increased proteolytic activity. Gate opening and binding interface protection are expected to have opposite impacts on RDU, so their combination in the same protein region would result in no significant changes of deuteration rates.”

Concerning the association we make between protection and affinity in the Results section, we understand that some of our statements may appear too strong. Accordingly, we made the following changes:

Line 223: *“The protection of PA28 γ activation loop (region 146-152) upon std20S binding was also statistically significant but notably less pronounced (corrected P-value = 1.49.10⁻⁸, sum of difference = -20%) (Fig. 6B and Table S4), which could indicate a greater affinity of the std20S for PA28 $\alpha\beta$ than for PA28 γ ”*. Although only stated as an hypothesis (*“could indicate”*), we decided to remove the end of the sentence (*“which could indicate a greater affinity of the std20S for PA28 $\alpha\beta$ than for PA28 γ ”*), which also allows us to comply with comment #10 of Reviewer #3.

Line 237: *“Most of the reduction in accessibility/flexibility that was observed on the PA28 chains upon binding to the std20S were dampened upon interaction with the i20S, suggesting a stronger interaction of the regulators with the std20S than the i20S (Fig. S9)”*. We displaced the sentence *“...suggesting a stronger interaction of the regulators with the std20S than the i20S (Fig. S9)”* from the Results section to the Discussion section.

Line 371: *“In this context, we think that protection to deuteration cannot be always correlated to relative binding affinities.”*

5. Page 4, line 108: Explain the workflow in more detail, refer to specifics outlined in Figure 1A.

Fig. 1 was modified and the workflow is now explained in more details in the Figure legend. We also added a description of our strategy line 112: *“We monitored the incorporation of deuterium for each sample from 0 to 30 min in triplicate experiments (Fig. 1B). For differential analyses, we compared the peptide relative deuterium uptakes (RDU) of the proteins alone vs. in presence of their binding partner. The peptides were considered significantly different when they presented a P-value of the ANOVA ≤ 0.01 after correction for multiple testing (Benjamini Hochberg) and 3 successive time points with an absolute difference between the two conditions > 4 times the experimental standard deviation observed in the data set (Fig. 1C) (see Methods for more detailed information). The peptide-level RDUs or differences of RDU were mapped to the protein sequences as described in Figure 1D: when several peptides covered the same region of a protein we kept only the RDU (or difference of RDU) of the shortest one. Finally, we mapped the RDUs to the protein 3D structures in order to visualize the regions the most accessible to solvent (Fig. 1E, left panel), or presenting significant differences of deuteration (Fig. 1E, right panel). For differential analysis, we colored the 3D structures with the sum of RDU differences between the two conditions (the values per time point are available in Fig. S4, S8 and S11). In this context, modification of solvent accessibility upon complex formation”*.

6. Page 4, line 109-112: Explain the consolidation and differential analyses in more detail, refer to specifics outlined in Figure 1B-E.

We believe that this comment is associated with comment #13. We apologize for not being clear in the first version of the manuscript. We have modified the Fig. 1 and we believe it is now clearer. The consolidation step is presented in the panel D, and more detailed in the legend:

“RDUs of the smallest peptides were mapped to the amino-acid portion they cover on the protein sequence. These were then presented as color gradients on heatmaps: light green and dark blue correspond to low and high RDU, respectively. The origin of RDU reported on the heatmap is indicated above the protein sequence with peptides represented as solid and dashed lines when their RDUs were reported or ignored, respectively. Residue numbers are indicated on the top.” Furthermore, we added a sentence at the beginning of the Results section that describes the workflow (see comment #5).

7. Along the manuscript changes in RDU are considered to be significant, however numbers are not provided to indicate significance. A table showing significant changes should be added where relevant.

The results of the statistical analysis are provided for each peptide in the Table S4:

- RDU differences across the kinetics (from 0.5 to 30 min)
- *P*-value of the ANOVA before and after correction for multiple testing (Benjamini Hochberg)
- The peptides passing the statistical thresholds are indicated in the last column

It is true that it can be time consuming to report back to this table while reading the manuscript so we added the RDU differences and corrected *P*-values for some of the peptides discussed in the main text. Due to space limitation, and because we believe that the main text would be hard to read if we added these information systematically for all the peptides commented, we were not exhaustive. We focused on a selected set of protein regions that seemed to be of high interest, we would be happy to be more exhaustive if requested.

8. No information is provided about the two antechambers. Is there more solvent exposure when the pore is open? Please discuss.

The reviewer is right to say that we did not use the term “antechamber”. Instead, we talked about the interface between the α -ring and β -ring interface (shown in **Fig. 2D/E**, **Fig. 4B/C** and in **Fig.7** as well), which is indeed not exactly the same, since the interface could refer only to the residues that are in contact, whereas the antechamber refers to the cavity itself. We thank the reviewer for spotting this and we added a new panel on **Fig. 4** precisely showing the cavities of the 20S proteasome and how the solvent accessibility of certain stretches increases upon incorporation of the immuno catalytic subunits. The figure legend was modified and the regions of the antechambers that are more deuterated in the

i20S vs. std20S are now described both in the figure (red arrows) and in the text as follows: ***“as well as the various regions from the antechambers that were more accessible in the i20S vs. std20S (98-99 in $\alpha 2$; 102-109 in $\alpha 3$; 128-142 in $\alpha 5$; 100-108 in $\alpha 6$; 95-105 in $\beta 3$; 94-102 in $\beta 4$, Fig. 4E).”***

9. The binding of 20S and 20Si to the PA28 regulators is very confusing as there is no obvious pattern. Therefore, supporting experiments are required to rule out technical issues.

This comment is related to comment #3 of Reviewer #3 and comment #1 of Reviewer #1, in which he/she asks for additional experiments to prove the activation of the 20S proteasome by the PA28 regulators.

To validate the complex formation, we checked that both PA28 regulators utilized in this study were able to activate the std20S proteasome using the same ratio as in the HDX-MS experiment (2-fold molar excess of PA28 regulator), as shown in our new Supplementary Fig. S8 and explained on line 214: ***“Before comparing the PA28 regulators alone and in complex with the std20S, we confirmed their ability to increase the proteasome activity in vitro. In our hands, a 2-fold molar excess of any of the PA28 regulators doubled the std20S chymotrypsin-like activity (P-value of a paired two-sided t-test: 0.0007 and 0.0013 for PA28 γ and PA28 $\alpha\beta$, respectively; Fig. S8 and Table S6)”***. Thus, the absence of significant protection of the std20S α subunit N-termini cannot be explained by a lack of PA28 activity/complex formation.

We are now also discussing this point on line 372 of the manuscript:

“The very limited decrease of RDU on the α -ring interface interacting with PA28 regulators can be explained by the resulting gate opening (RDU increase) known to induce its increased proteolytic activity. Gate opening and binding interface protection are expected to have opposite impacts on RDU, so their combination in the same protein region would result in no significant changes of deuteration rates.”

10. The suggested higher affinity of 20S to PA28ab than to PA28g should be validated by an additional methods.

We agree with the Reviewer #3 and decided to remove this sentence (l.22: ***“which could indicate a greater affinity of the std20S for PA28 $\alpha\beta$ than for PA28 γ ”***) from the text, in order to avoid any over-interpretation of our data. This also contributes to answer to comment #4 or Reviewer #3 (link between level of protection and affinity).

11. Prove activity and purity of the purchased complexes.

As explained above (Reviewer #1, comment #1) we confirmed the activity of the commercial complexes. This result is commented on line 214: ***“using a 2-fold molar***

excess of PA28, that was enough to activate the 20S proteasome” and included as **Supplementary Fig. S8** and **Table S6**. The numerotation of the supplementary figures and tables was modified accordingly and the details of the proteasome activity test are now described in the Methods section.

Concerning the purity of the purchased complexes, we recently used Top-Down MS to establish and publish the 3D proteoform footprints of the std20S and i20S from Enzo⁶¹. These results showed a very good purity for the std20S (containing only std20S catalytic subunit) and a slight contamination of the i20S from Enzo. Other commercial i20S from a different vendor (Boston Biochem) also revealed the presence of traces of std20S (Gersch et al, 2015).

We added a comment regarding this explanation in line 328: *“As commented before, the differential analysis of the catalytic subunits is impossible due to their sequence differences (the peptides resulting from the pepsin digestion are different). The comparison of their deuteration profiles was further hindered by the limited sequence coverage of the i20S. This was partly due to the presence of residual traces of std20S present in this sample⁶¹”*.

As explained in the first comment of Reviewer #2, in the case of co-existing conformations, HDX-MS provides an average accessibility. This means that the presence of traces of std20S in the i20S sample, when compared to pure std20S, only attenuates the difference observed between the two samples.

12. Page 6, lines 153-155: How much more deuterated is T20 and T22 of B1/B1i? State the numbers as by eye in the figure it doesn't look very different. Is it a significant difference? Same goes for the S1 pocket of B5/B5i, list the relevant amino acids (A20/V31?) and their RDU in the text. Can the conclusion for more flexibility in the active sites/S pockets really be drawn for a small difference in RDU for one or two residues? If yes then it need to be clearly justified.

The section title is misleading, as stated in the text the direct comparison between the STD and i20S catalytic subunits was not possible due to poor sequence coverage, therefore no statement can be made on any conformational difference between the two proteasomes. Thus, the concluding sentence is also misleading, as there is not enough of a difference in their flexibility. If this is not the case then explain in more detail.

We thank Reviewer #3 for his/her critical comments. We agree that our conclusions on the comparison of the catalytic domains of the std20S and i20S were too strong, and we apologize. We tuned down our statements in the main text and merged the paragraph dedicated to the comparison of the std vs. i20S catalytic domains with the following paragraph: “An inner-to-outer conformational change upon substitution of standard to immuno-subunits”.

13. Figure 1: In general the figure is very busy, can it be simplified?

We followed Reviewer #3's suggestion and simplified the **Fig.1**. We added a panel (new panel A) that outlines (and hopefully clarifies) the differences between our two approaches: analysis of proteins alone or differential analysis of proteins alone vs. in complex. We also described the workflow in more detail in the legend. We hope this is now clearer.

A: Describe the workflow in more detail in the legend.

The workflow is now described in more detail in the legend.

B: "Protein sequence" should refer to the amino acid letter sequence that is directly above the heatmap, "residue number" should be labelled above the arrow

These are now described in the legend.

Describe in the legend what the heatmap colours mean i.e. light blue is less RDU, dark blue is more RDU? In general across all the figures RDU changes from a percentage to a decimal number.

The color-codes are now described in the legend. We also homogenized the RDU notations and only provide percentages.

D: Describe in the legend the statistical analysis outlined

The statistical analysis is now described in the legend.

Is there correlation between the long and small peptides? Explain the rationale for choosing only the smaller peptides.

The same region of a protein can be monitored by several peptides of different length. Their total RDU kinetics are different depending on the solvent accessibility of the peptide sequence in the structured protein. There are several strategies utilized in the HDX community to "translate" RDU values from peptides to protein regions. We chose to report the RDU of the shortest peptides because it is the value that best represents the accessibility of this protein region, and it was in our hands the most robust strategy.

Figure 1 (below) presents a comparison of the consolidation outputs using 3 different strategies:

- RDU of the shortest peptide covering a given residue
- mean value of the peptides covering the same region of the protein (weighted by $1/(\text{peptide length})$)
- subtraction-based consolidation described in (Pascal et al, 2009)

For each protein of this study, we calculated the Pearson correlation between the RDU values of each residue resulting from two consolidation strategies. A higher correlation corresponds to more similarity between the two consolidation strategies compared. In these data, using the weighted mean or the shortest peptide resulted in very similar RDU kinetics (high correlation in the first pairwise comparison: 1st column of the figure). The output of a subtraction-based consolidation did not correlate as well with the two other methods. Overall, the strategy using the shortest peptide is in our opinion the most straightforward and the one that agrees the best with the two other strategies. In any case, we provide the peptide coverage as well as the information of which peptide RDU is reported in the deuteration heatmaps in the supplementary figures so that the reader can make his/her own opinion on the data.

Fig.1: Comparison of consolidated values obtained with three strategies utilized to map peptide-level RDU to protein sequences. We mapped the peptide RDUs to protein sequences using: (i) the mean values of overlapping peptide weighted by 1/(peptide length) (“weighted mean”); (ii) the value of the shortest peptide covering the protein region (“shortest pep.”); or (iii) a subtraction-based consolidation described in (Pascal et al, 2009). The proteins are ordered by hierarchical clustering. For this figure, we only used the data from proteins alone in solution.

14. Figure 2: The regions highlighted with red arrows in C-F are not specifically mentioned in the text as stated in the legend. Describe their significance in the text or in the legend.

We thank the reviewer for spotting this mistake and apologize for it. This is partly due to numerous modifications of the draft and our efforts to reduce the size characters to comply with Nat. Commun. Editorial policy. We now mention in the text all the regions that highlighted in red in **Fig. 2** and have removed from **Fig. 2** any region that is not mentioned in the text.

B: What is the red circle highlighting? Not described in the legend

This region highlights a dynamic patch at the interface between the β -rings that was commented in a previous version of the text but was removed due to space limitations. We thank the referee for spotting this. The red circle is now removed from **Fig. 2B**.

15. Page 5, lines 128-133: The patches on the interface that correspond to the pockets for the RPT tails should be highlighted in Figure 2 for easy identification.

We thank the reviewer for this suggestion. As also requested by Reviewer#1 (comment #2), we circled in white the 7 α -pockets, as stated in the legend: *“The α -pockets (docking sites for the C-termini of PA28 regulators) are circled in white in (C).”*

Also recommended to elaborate on the regions highlighted in with red arrows in the figure, what is their relevance?

As mentioned in the previous comment, we now state in the text all the regions that are highlighted in red in **Fig. 2**.

16. Figure 3: Why not include the differential analysis of the catalytic subunits here to make the differences between the STD and i20S more obvious? Or expand Figure 4 to highlight the differences.

Unfortunately, it is not possible to directly compare the RDU profiles of the catalytic subunits of the std20S with those of the i20S because their sequences are different. The peptides resulting from their pepsin digestion are not the same and cannot be matched for comparison. For this reason, we did not calculate RDU differences, or perform statistics on these subunits, which explains why these are black in **Fig. 4**. Instead, we presented in **Fig. 3** an alignment of the independent measurements obtained with the two 20S proteasomes. We believe that this representation (with peptide coverage and color-coded RDU) is the best way to visualize the data we gathered on the catalytic domains. It would be possible to merge **Fig. 3** and **Fig. 4**, but we thought that **Fig. 3** was already very busy so we kept them separated.

We added more precision in the second paragraph of the Results section, to clarify why we cannot directly compare the catalytic subunits of the std20S vs. i20S proteasomes with HDX-MS: *“The direct comparison of RDUs between the standard and immuno-catalytic subunits ($\beta 1/\beta 1i$, $\beta 2/\beta 2i$ and $\beta 5/\beta 5i$) was not possible due to their sequence differences”*.

17. Figure S2, S6 and S10: Provide the same y axis for all graphs to allow a quicker comparison between the RDUs of each peptide possible.

The same y-axis is now used for all the peptides of each subunit in **Fig. S2**, **Fig. S6** and **Fig. S10** (now **Fig. S11**).

18. Page 7, line 168: As mentioned in the previous section, the differences in the catalytic and substrate pockets between the STD and i20S were not significant, so how can this translate to significant conformational changes in the non-catalytic subunits?

As mentioned previously, the poor sequence coverage and difference in sequence of the peptides does not allow us to directly compare the deuteration of the standard and immuno subunits. It doesn't mean that there is no difference, but that we cannot fully appreciate them, if present. Even if very minor, these differences could very well propagate and be amplified to the contiguous subunits on which we could perform statistical analysis. We have modified the text accordingly in the second paragraph of the Results section: *“The direct comparison of RDUs between the standard and immuno-catalytic subunits ($\beta 1/\beta 1i$, $\beta 2/\beta 2i$ and $\beta 5/\beta 5i$) was not possible due to their sequence differences. Nevertheless, we could acquire the deuteration profiles of the peptides ADSRATAGAY (16-25) of $\beta 5$ and AVDSRASAGSY (15-25) of $\beta 5i$ that cover the same portion of the two subunits. Their difference of deuteration profiles indicate that these residues are more flexible in std20S than in the i20S (Fig. 3D and Fig. S2). Allosteric differences beyond the catalytic subunits were shown in the prokaryotic std and i20S analysed by NMR³⁴. These were not observed in mouse crystallographic structures³¹, maybe due to crystal packing. In order to confirm or infirm this inner-to-outer allosteric change in human, we compared the RDUs of all the non-catalytic subunits of the i20S (Fig. S2, S3 and Table S2) with those of the std20S (as presented Fig. 1). The introduction of immuno-catalytic subunits resulted in significant conformational changes on the non-catalytic 20S subunits.”*

19. Figure 5:

A: Point out the missing residues 15-31 in the structure in 5A, is this the dashed line?

We believe that this comment can be linked to Reviewer's #1 comment #4. Here, “15-31” meant that the missing apical loops were encompassing 15 to 31 residues, depending on which PA28 regulator we are talking about (the length of the loop and the length of the missing residues in the PDB files varies from one monomer to another). We are sorry for this misunderstanding and replaced “15-31 residues” by **“15 to 31 residues”**.

The apical loops are now also represented as dashed lines in Fig.5B. This is clearly stated in the figure legend: *“The residues corresponding to the missing apical loops (dashed lines) are indicated in (B).”*

Include a cartoon representation of the entire heteroheptamer, showing the position of the apical loops

The cartoon representation of the heptamer was added as a new panel (now Fig.5C). The figure legend was modified accordingly: *“(C) Structure of the heptameric complex PA28 α 4 β 3 (PDB: 5MX5). Only 4 apical loops out of 7 are represented, for the sake of clarity”*.

B: The RDU scale is 0-250%, but in Fig S7, from which these structures are based, the RDU goes from 0-100% (0.0-1.0)?

This is right. In Fig.5B/D/E, as stated in the Figure legend, we represented the sum of the RDU obtained at each timepoint to visualise global changes over the time course. Fig.S7 present the same proteins, but with all measured values across the time course (one time point per row) The two figures do not present the same values, hence the difference in the scales of these two figures.

By eye the blue color of the RDUs don't match up to what is represented in the structures, in B-D.

We used the gnbu-5 color palette in ChimeraX for all the structures represented in Fig.5B/D/E. The only difference between Fig.5B and Fig.5D/E is the representation (from cartoon to surface), no change was made to the color scale. Here again we would like to remind that in Fig.5 we represented the sum of the RDUs, whereas Fig.S7 represents the RDU obtained for each individual time point (so the scale and colors are different between these two figures).

The .csx sessions used for the direct 3D visualization of the deuteration uptakes in ChimeraX are available in our Pride Repository.

C-D: Include arrows showing direction of rotation between the structures

Arrows have been added to Fig.5D-E, as suggested by the reviewer.

D: State in legend that this is a homology model.

This is now stated in the Fig.5E legend (line 701: *“E, homology model”*).

20. Page 8, line 211-212: Elaborate on the significance of the activation loop of PA28B being less deuterated, as this is the title of the section it should be explained in more detail. Why is this important?

The relevance of the results obtained on PA28 activation loop is further detailed in the Discussion section as follows: *“Although present in the three PA28 subtypes, the activation loop protection was stronger in PA28 α and PA28 γ vs. PA28 β ; most probably explaining why “the affinity of the proteasome towards the PA28 $\alpha\beta$ complex is about two orders of magnitude higher than towards the homomeric PA28 α and PA28 β complexes⁶². Indeed, several studies showed that both PA28 α and PA28 β were required for an optimum activation of the 20S by PA28 $\alpha\beta$ ^{32,33}.”*

We agree with Reviewer #3 that, since this is the title of this Results paragraph, this part deserves more emphasis here. We thus added the following sentence at the end of this paragraph: *“The reason why eukaryotic homoheptameric PA28 regulators have evolved towards more complex heteroheptameric PA28 $\alpha\beta$ is not well understood. Given the overall similar tridimensional structures of PA28 $\alpha\beta$ and PA28 γ , the poorer flexibility observed on PA28 β activation loop could, at least partially, explain their different functions.”*

21. Figure 6: State in the legend that the color coding indicates that blue is less accessible in the complex and red is more accessible in the complex.

This is now included in the legend of **Fig.6**.

Include arrows to show direction of rotation between the images.

Arrows have been added to **Fig.6**, as suggested by the reviewer.

Change the colors of the black arrows pointing at the activation loops, they are hard to see.

The arrows pointing at the activation loops are now in red, in order to increase visibility. We thank the reviewer for suggesting this.

Include the differential analysis images of the PA28 with the i20S

The corresponding structures were added to **Fig. 6** (panels C and D), as suggested by the reviewer. The **Fig. 6** title and legend were modified accordingly.

In Fig S8 it is clear that the effect is the opposite, i.e. the PA28 subunits become more accessible upon i20S binding. This is briefly mentioned on page 9, lines 235-238, elaborate on this, it is interesting.

We think that there is a misunderstanding here. The effect of i20S binding to PA28 is shown on pages 2 (for PA28 α), 5 (for PA28 β) and 8 (for PA28 γ) of the **Fig. S8** (now **Fig.**

S9). As stated in the text, and now shown on **Fig.6C-D**, the effect is similar to the one observed with the std20S, although dampened.

We think that Reviewer #3 mentions pages 3, 6 and 9 of the **Fig. S8**, that correspond to the differential analysis of PA28+std20S vs. PA28+i20S (and not PA28 alone vs. PA28+20S).

The colours are different in this differential analysis (basically more red), because they represent different things:

- In the PA28 alone vs. PA28+20S: blue means more protected in the complex and red means more exposed in the complex
- In the PA28+std20S vs. PA28+i20S: blue means more protected with the i20S (than with the std20S) and red means more protected with the std20S (than with the i20S).

This point is now better explained at the end of the 4th paragraph of the Results section: **“We then compared the impact of std20S vs. i20S binding on PA28 solvent accessibility using the statistical strategy presented in Fig. 1C (comparison referred to as “PA28+std20S vs. PA28+i20S” in the figures)”**.

22. Page 9, line 220: These listed residue regions are not highlighted in the figure, and in the figure there are residue regions highlighted in red that are not mentioned in the text?

We thank the reviewer for spotting this mistake and apologize for it. This is partly due to numerous modifications of the draft and our efforts to reduce the size characters to comply with Nat. Commun. Editorial policy. We have now amended **Fig.6** to remove any region that is not mentioned in the text and highlight in red all the regions that are mentioned in the text. We also added a view from the inside of the regulators to better illustrate the regions that are protected (especially the constriction sites that are now visible).

23. Page 9, line 225: State the residue numbering of the kinks. Why are they indicated?

The residue numbering of the kinks is now stated, as requested. We think that they are playing an important role in the activation mechanism of the 20S by the PA28 regulators, as explained in the discussion and resumed in **Fig.8**.

24. Figure 7: This is one of the most important results but it not clear at all. In general, the description in the text of the results in Figure 7 is extremely difficult to follow and is confusing. Perhaps representing the data in a different more manageable way is warranted. For example, include larger labels for the columns of structures at the top of the figure, indicating which columns correspond to STD and i20S, combined with which PA28 regulators. Explain in the legend what the red circles are referring to.

We thank the Reviewer #3 for these suggestions that have now been implemented in a new version of **Fig.7**. We hope that it is now easier to follow and less confusing for the reader.

25. Page 10, line 247-249: State the differences in deltaRDU calculated for the C-ter of a3 depicted in Fig7A and Fig 4D, by eye the delta RDU looks like it indicates a strongly protected region in both analyses.

As suggested, we added the corrected *P*-values and Sum of RDU differences for this peptide in all three cases in line 247-249.

26. Page 10, line 251: the regions in 7A that are being referred to are blue from the analysis but circled in red? If there are circles in the figure refer to them, or leave them out.

As mentioned by the Reviewer #3 (comment #24), the previous version of **Fig.7** was hard to follow. We decided not to overcrowd this **Fig.7** with a detailed labeling of all the regions that are significantly more deuterated (colored in red) or less deuterated (colored in blue) upon complex formation and opted to circle in red only the ones (blue or red) that are now discussed in the text. Furthermore, as further detailed in comments #30/31/32/33 of Reviewer #3, we recognize that these regions of interest need to be better referred to when there are different circles in a single panel. We thus decided, in that case (several circles in a single panel), to add the subunit it belongs to in the text.

27. Discussion: In general, the discussion reads more like a summary of the results section, and not a discussion of the results in the context of the literature. A summary of the results is definitely warranted, although perhaps this can be done briefly at the beginning of the discussion section, and not take up the majority of it.

We are sorry if the previous version of the discussion appeared “weak” to Reviewer #3, although it already contextualised our results (state of the art of HDX-MS analysis of large complexes, structural comparison of 50 proteasome-related structures, the latest PA200 cryo-EM and mouse and human 20S structures, the prokaryotic PA28-20S complex, the Insulin-Degrading-Enzyme binding partner...). We added in the new draft a reference to a very recent cryo-EM structure of PA28 $\alpha\beta$ in complex with bovine spleen i20S: ***“in line with a recent cryo-EM structure that showed a partial opening of the bovine i20S gate in complex with human PA28 $\alpha\beta$ ”***⁶⁵. Overall, we believe that the whole manuscript, including the Discussion section was greatly improved thanks to all the insights we received from the 3 Reviewers. We hope that this new version of the manuscript now suits the Reviewer’s expectations.

28. Figure 8: The legend needs to be much more descriptive. This is the model of your major findings, explain in much greater detail.

We made a new version of Fig.8 to answer to comments #28/38/40-42 of Reviewer #3. We did our best to improve the representation of our main findings and adapted the legend accordingly.

A: Colors in the text are referred to as orange and purple?

We thank the reviewer for spotting his confusion, which comes from an old version of the figure.

The term “purple” was replaced by “**blue**” on line 313.

The term “orange” was replaced by “**red**” on line 328 and 329.

B: The text describes this as showing stabilization of the loops, this isn't clear in the figure.

We apologize for this. The stabilisation of the regulator activation loops upon binding to the proteasome was shown in **Fig.8C** and not in **Fig.8B**. It was represented by a decrease of deuteration (-). It has now changed with the new version of the figures and was corrected in the main text.

Also, what are the + and - symbols representing?

The meaning of these symbols is now explained in the **Fig.8** legend.

C: What are the – symbols representing?

The meaning of this symbol is now explained in the **Fig.8** legend.

29. Methods: Page 16, lines 396-398: Include catalog numbers for the listed reagents.

The catalog numbers were included for the listed reagents.

30. Page 10, line 259: Which red circles in 7C are being referred to? There are 3...

As mentioned above (comment #26) we now state in the text, the subunit(s) corresponding to the red circle in question, in this case: “**Fig. 7C, red circles on $\beta 5$, $\beta 6$** ”.

31. Page 10, line 262: There are no blue circles in 7C, only red circles. Clarify which region is being referred to.

We thank the Reviewer #3 for spotting this. The sentence “**circled in blue**” was replaced by “**red circle on $\beta 1$** ”.

32. Page 11, line 266-267: Stated that the binding of PA28g only affected $\alpha 2$ and $\alpha 3$ Nterminals of the i20S, and yet in 7P it clearly looks like the $\alpha 1$ Nter is also destabilized, even more than the $\alpha 2$?

We double checked this part and clarified our results as follows: the N-terminal part of $\alpha 3$ (in the i20S) is not affected by PA28 γ . However, the region that the Reviewer #3

mentioned here does not belong to $\alpha 1$ but to $\alpha 7$ (region 2-9, as seen in Supplementary **Fig.S12** and **S13**). The red circle encompassing regions from both $\alpha 1$ and $\alpha 7$ in **Fig.7P** was clearly misleading. We have thus replaced “ **$\alpha 2$ and $\alpha 3$ N-ter**” by “ **$\alpha 2$ and $\alpha 7$ N-ter**” on line 266 and used a larger red circle for all the N-ter parts of the α -ring, as in **Fig. 7A/K**.

33. Page 11, line 273: Where in 7F is the $\alpha 2$ destabilization? Why is it not circled like the other regions being discussed in 7H?

The region 101-119 was indeed not circled in **Fig.7** and we thank the Reviewer #3 for spotting this. This region is actually visible on the other side of the ring (**Fig.7G**) and we have now circled it in red as mentioned in the text: “***This seemed to induce the PA28 γ -specific destabilization of $\alpha 2$ (101-119: corrected P -value = $4.03 \cdot 10^{-5}$, sum of difference = 13%, Fig. 7G, red circle on $\alpha 2$)***”.

34. Page 11, line 282-283: Include the RDU numbers to demonstrate the significant changes upon PA28ab binding as by eye it doesn't look convincing. Perhaps a small table with the relevant numbers in the figures throughout the manuscript that are significant would help.

Following comments #7 and #36 of Reviewer #3, we added the sum of RDU differences and corrected P -values for these peptides in line 282-290 (and for others throughout the text).

We agree with Reviewer #3 that being able to quickly find in the Tables the corrected P -values (“pBH”) and Sum of RDU differences for each peptide that is regulated would be a plus. We thus created a new Supplementary **Table S4**. Unfortunately, this table can not be small as requested by Reviewer #3, since we identified 1,737 regulated peptides in our data set (considering all the differential analysis described in the text), out of 7,249 peptides.

35. Page 11, line 284: The B1i active site in 7N does not look significantly affected, in fact most of the active site circled in orange is gray.

In line 284, we refer to the peptide 10-17 of $\beta 1i$, encompassing the catalytic residue D17, that is significantly more deuterated in the presence of PA28 $\alpha\beta$ (corrected P -value = $3.89 \cdot 10^{-3}$, sum of RDU differences = 14%). The reviewer is right to say that this peptide is not visible on **Fig.7N**, because it is masked by peptides 18-24 and 162-175. We thus modified the text to cite **Fig.S11-S12** instead and added the statistics obtained for peptide 10-17.

This question was also raised by Reviewer #1 (comment #6).

36. Page 12, line 289-290: What are the numbers to say there is a significant decrease? Is the B2i N-terminal not visible in Fig 7S? State this if this is the case.

Following comments #7 and #34 of Reviewer #3, we added the sum of RDU differences and corrected *P*-values for these peptides in line 282-290.

37. Page 12, line 312: First mention of residues 238-261, where in the results section is this described?

Following comment #2 of Reviewer #1, we now cite these residues in line 128 of the results section. Additionally, we also cite them in line 248.

38. Page 13, line 313: There is no purple coloring in Figure 8.

The term “purple” was replaced by “*blue*” on line 313.

39. Page 13, line 323-326: Elaborate on the comparison with Pf20S, why is this species being mentioned? Is this one of the few structures of 20S bound to PA28?

We wanted to see here if the α 3 C-ter was somehow stabilized or interacting with PA28 homologues.

As stated line 202: *“The sequence of Pf is used as a comparison here because it is part of the very few structures of PA28 regulator homologues available in the PDB with close homology to PA28 γ and whose structure was recently solved in complex with the 20S²⁴”*.

40. Page 13, line 328: There is no orange coloring in Figure 8.

The term “orange” was replaced by “*red*” on line 328.

41. Page 13, line 331: There are no orange arrows in Figure 8.

The term “orange” was replaced by “*red*” on line 329.

42. Page 14, line 359: Figure 8B doesn't seem to be showing stabilization of the loops?

We apologize for this mistake. The stabilisation of the regulator activation loops upon binding to the proteasome was shown **Fig.8C** and not **Fig.8B**. It was represented by a decrease of deuteration (-). It has now changed with the new version of the figures and was corrected in the main text.

Additional comments

43. Page 2, lines 42-48: Include references for the UPS and its functions.

Four recent references were added:

1. Collins, G. A. & Goldberg, A. L. The Logic of the 26S Proteasome. *Cell* 169, 792–806 (2017).
2. Bard, J. A. M. et al, Structure and Function of the 26S Proteasome. *Annu. Rev. Biochem.* 87, 697–724 (2018)
3. Groettrup, M., Kirk, C. J. & Basler, M. Proteasomes in immune cells: more than peptide producers? *Nat. Rev. Immunol.* 10, 73–78 (2010).
7. Coux, O., Zieba, B. A. & Meiners, S. The Proteasome System in Health and Disease. in *Proteostasis and Disease : From Basic Mechanisms to Clinics* (eds. Barrio, R., Sutherland, J. D. & Rodriguez, M. S.) 55–100 (Springer International Publishing, 2020). doi:10.1007/978-3-030-38266-7_3

44. Page2, lines 51-55: Include more detail about the catalytic subunits i.e. what are their different enzymatic activities

The enzymatic activities of each catalytic subunit are now detailed line 51-53: *“The main mechanism regulating the proteasome activity is the substitution of the catalytic subunits $\beta 1$, $\beta 2$ and $\beta 5$, respectively harboring caspase-like, trypsin-like and chymotrypsin-like activities, and constituting the standard 20S (std20S) with other subunits.”*

45. Page 3, line 72: Other citations needed for additional PIPs beyond Ecm29

The sentence on line 72 is about PIPs (including Ecm29) whose structures have been modelled with the help of cross-linking, which are not numerous. We only found one additional example on Ubp6³⁰. The corresponding reference was added:

30. Aufderheide, A. et al, Structural characterization of the interaction of Ubp6 with the 26S proteasome. *Proc. Natl. Acad. Sci.* 112, 8626–8631 (2015).

The other references on PIPs are detailed on line 60 and a new reference on PI31¹⁶ was also added:

16. Minis, A. et al, The proteasome regulator PI31 is required for protein homeostasis, synapse maintenance, and neuronal survival in mice. *Proc. Natl. Acad. Sci.* 116, 24639–24650 (2019).

46. Page 4, line 90: Correct P28 to PA28

The text has been modified accordingly.

47. Page 4, line 90-92: Elaborate on the HDX-MS method, a more detailed explanation is required. Also include references for the method.

A more detailed description of the HDX-MS methods, with appropriate references was added to the text: *“This method, developed in the 90s, enables the detection of conformational differences between two protein samples. Briefly, the proteins or protein complexes of interest are diluted in a deuterated buffer and hydrogen atoms from the peptide bonds are exchanged with the deuterium atoms of the solution. The rate of exchange of each amide hydrogen depends on its solvent accessibility and involvement in the stabilization of secondary structures: regions that are relatively more solvent-accessible or flexible present a higher deuteration rate than the ones that are hidden in the core of the protein or rigid. This expanding method can thus be used to study protein dynamics³⁸, compare protein conformations (in different buffer conditions, after mutation, ligand binding)³⁹, or to identify protein-protein or protein-ligand binding interfaces^{40,41}.”*

48. Page 6, lines 159-161: ‘mice’ should be ‘mouse’. This sentence is confusing. Reword to make it clearer.

The text was modified (mouse instead of mice) and the sentence was split into two sentences to make it clearer: *“Although mouse i20S and std20S possess very similar $\beta 2$ and $\beta 2i$ substrate binding channels, there are structural differences between their standard and immuno-catalytic subunits $\beta 1$ and $\beta 5$. The S1 and S3 substrate pockets are smaller in $\beta 1i$ vs. $\beta 1$, $\beta 5i$ possesses smaller S2 and S3 pockets, and the S1 pocket is larger in $\beta 5i$ than in $\beta 5^{25}$ ”.*

49. Page 6, lines 147: ‘remaining’ should be ‘remainder’

The text has been modified accordingly.

50. Page 8, line 190: “An activation loop is less dynamic in PA28B”

We propose to replace the title of this paragraph by *“The activation loop of PA28 β is less dynamic/accessible than PA28 α/γ s”*. This way, we also comply with Reviewer #3’s comment #1 concerning the interpretation of a reduced deuteration: it can indeed be due to a reduced dynamics but also reduced accessibility.

51. Page 8, line 200: Define Pf, and explain why this species sequence is being used for comparison

The acronym of *Plasmodium falciparum* is now fully detailed on line 200. As stated line 202 *“The sequence of Pf is used as a comparison here because it is part of the very few structures of PA28 regulator homologues available in the PDB with close homology to PA28 γ and whose structure was recently solved in complex with the 20S²⁴”.*

The term *Plasmodium falciparum* was replaced by *Pf* on line 324.

52. Page 8, line 202: Include a more recent reference for the SWISS-MODEL server.

The reference was replaced by the most recent one:

52. Waterhouse, A. et al, SWISS-MODEL: homology modelling of protein structures and complexes. *Nucleic Acids Res.* 46, W296–W303 (2018).

53. Page 10, line 252: ‘remaining’ should be ‘remainder’

The text has been modified accordingly.

54. Page 10, line 255: State that all the regions being discussed in 7A are circled in red.

The text has been modified accordingly on line 255 and in the **Fig.7 legend**.

55. Page 10, line 258: “...and is in close vicinity to the B5 interface, which is also more dynamic”

The text has been modified accordingly.

56. Page 13, line 335: PGPH activity should be referred as caspase-like activity. Also “than the one of” should read “than that of”

The text has been modified accordingly.

References used for the answer to Reviewers:

Deng B, Lento C, Wilson DJ. (2016) Hydrogen deuterium exchange mass spectrometry in biopharmaceutical discovery and development - A review. *Anal Chim Acta.* 940:8-20.

Fang J, Nevin P, Kairys V, Venclovas Č, Engen JR, Beuning PJ. (2014) Conformational analysis of processivity clamps in solution demonstrates that tertiary structure does not correlate with protein dynamics. *Structure.* 22(4):572-581.

Faull SV, Lau AMC, Martens C, Ahdash Z, Hansen K, Yebenes H, Schmidt C, Beuron F, Cronin NB, Morris EP, Politis A. (2019) Structural basis of Cullin 2 RING E3 ligase regulation by the COP9 signalosome. *Nat Commun.* 10(1):3814.

Gersch M, Hackl MW, Dubiella C, Dobrinevski A, Groll M, Sieber SA. (2015) A mass spectrometry platform for a streamlined investigation of proteasome integrity, posttranslational modifications, and inhibitor binding. *Chem Biol.* 22(3):404-11.

Goswami D, Tuske S, Pascal BD, Bauman JD, Patel D, Arnold E, Griffin PR. (2015) Differential isotopic enrichment to facilitate characterization of asymmetric multimeric proteins using hydrogen/deuterium exchange mass spectrometry. *Anal Chem.* 7;87(7):4015-4022.

- Hvidt A, Nielsen SO. (1966) Hydrogen exchange in proteins. *Adv Protein Chem.* 21:287-386.
- Kaltashov IA, Eyles SJ. (2002) Crossing the phase boundary to study protein dynamics and function: combination of amide hydrogen exchange in solution and ion fragmentation in the gas phase. *J Mass Spectrom.* 37(6):557-65.
- Li X, Eyles SJ, Thompson LK. (2019) Hydrogen exchange of chemoreceptors in functional complexes suggests protein stabilization mediates long-range allosteric coupling. *J Biol Chem.* 294(44):16062-16079.
- Pascal BD, Chalmers MJ, Busby SA, Griffin PR. (2009) HD desktop: an integrated platform for the analysis and visualization of H/D exchange data. *J Am Soc Mass Spectrom.* 20(4):601-10.
- Skinner JJ, Lim WK, Bédard S, Black BE, Englander SW. (2012) Protein hydrogen exchange: testing current models. *Protein Sci.* 21(7):987-95.
- Wales TE, Engen JR. (2006) Partial unfolding of diverse SH3 domains on a wide timescale. *J Mol Biol.* 357(5):1592-604.

We are grateful to reviewers whose concerns and criticisms helped us to provide a significantly improved version of the manuscript and hope that they will now find it appropriate for Publication in Nature Communications

REVIEWERS' COMMENTS

Reviewer #1 (Remarks to the Author):

The revised manuscript by Lesne and colleagues is improved over the original manuscript. The authors made an effort to address the criticisms that were made during the first review. This reviewer has just one more question about the relationship between the proteolytic activities in new Fig. S8 and the RDUs in Figs. 7A and 7F.

Although no significant difference was observed in the proteolytic activities between std20S + PA28 γ and std20S + PA28 $\alpha\beta$ in Fig. S8, the effects on the std20S α subunit N-ter are quite different between PA28 $\alpha\beta$ and PA28 γ in Figs 7A and 7F. If there are any possible explanations about these seemingly inconsistent data, this reviewer would like the authors to discuss this issue in the revised manuscript.

Reviewer #2 (Remarks to the Author):

The authors have gone to great lengths to answer the reviewers' concerns and I am satisfied that my comments were addressed. I congratulate the authors for this impressive piece of work.
Frank Sobott

Reviewer #3 (Remarks to the Author):

The authors answered my concerns, I think the paper is suitable for publication

REVIEWER COMMENTS

Reviewer #1 (Remarks to the Author):

The revised manuscript by Lesne and colleagues is improved over the original manuscript. The authors made an effort to address the criticisms that were made during the first review. This reviewer has just one more question about the relationship between the proteolytic activities in new Fig. S8 and the RDUs in Figs. 7A and 7F.

Although no significant difference was observed in the proteolytic activities between std20S + PA28 γ and std20S + PA28 α 13 in Fig. S8, the effects on the std20S α subunit N-ter are quite different between PA28 α 13 and PA28 γ in Figs 7A and 7F. If there are any possible explanations about these seemingly inconsistent data, this reviewer would like the authors to discuss this issue in the revised manuscript.

We thank Reviewer #1 for his/her comment. We think that different activators could mediate proteasome activation to a similar level with different molecular processes. Unfortunately, we were not able to further explore this phenomenon with HDX-MS. We now discuss potential explanations for our results in the Discussion section: ***“We showed that both PA28 regulators activate the chymotrypsin-like activity of the std20S to a similar extent in our in vitro assay (Supplementary Fig. 2). However, the N-ter portions of the std20S α subunits (entrance of the channel) were significantly more deuterated when interacting with PA28 $\alpha\beta$ whereas this increase was not significant with PA28 γ (Fig. 7A/F and Supplementary Data 3), suggesting that their allosteric mechanisms of activation are different. This could alternatively be explained by a shorter dwell-time at the surface of the std20S or by differences in binding stoichiometry. More experiments are needed to fully understand these allosteric mechanisms of activation.”***

Reviewer #2 (Remarks to the Author):

The authors have gone to great lengths to answer the reviewers' concerns and I am satisfied that my comments were addressed. I congratulate the authors for this impressive piece of work.

Frank Sobott

Reviewer #3 (Remarks to the Author):

The authors answered my concerns, I think the paper is suitable for publication.